

# Model design and parameter optimization of CNN for side-channel cryptanalysis

Yun Lin Liu, Yan Kai Chen, Wei Xiong Li and Yang Zhang

Center of Equipment Simulation Training, Shijiazhuang Campus of the Army Engineering University, Shijiazhuang, Hebei, China

## ABSTRACT

**Background:** The side-channel cryptanalysis method based on convolutional neural network (CNNSCA) can effectively carry out cryptographic attacks. The CNNSCA network models that achieve cryptanalysis mainly include CNNSCA based on the VGG variant (VGG-CNNSCA) and CNNSCA based on the Alexnet variant (Alex-CNNSCA). The learning ability and cryptanalysis performance of these CNNSCA models are not optimal, and the trained model has low accuracy, too long training time, and takes up more computing resources. In order to improve the overall performance of CNNSCA, the paper will improve CNNSCA model design and hyperparameter optimization.

**Methods:** The paper first studied the CNN architecture composition in the SCA application scenario, and derives the calculation process of the CNN core algorithm for side-channel leakage of one-dimensional data. Secondly, a new basic model of CNNSCA was designed by comprehensively using the advantages of VGG-CNNSCA model classification and fitting efficiency and Alex-CNNSCA model occupying less computing resources, in order to better reduce the gradient dispersion problem of error back propagation in deep networks, the SE (Squeeze-and-Excitation) module is newly embedded in this basic model, this module is used for the first time in the CNNSCA model, which forms a new idea for the design of the CNNSCA model. Then apply this basic model to a known first-order masked dataset from the side-channel leak public database (ASCAD). In this application scenario, according to the model design rules and actual experimental results, exclude non-essential experimental parameters. Optimize the various hyperparameters of the basic model in the most objective experimental parameter interval to improve its cryptanalysis performance, which results in a hyper-parameter optimization scheme and a final benchmark for the determination of hyper-parameters.

**Results:** Finally, a new CNNSCA model optimized architecture for attacking unprotected encryption devices is obtained—CNNSCAnew. Through comparative experiments, CNNSCAnew's guessing entropy evaluation results converged to 61. From model training to successful recovery of the key, the total time spent was shortened to about 30 min, and we obtained better performance than other CNNSCA models.

Corresponding author
Yun Lin Liu, llyun324@163.com

## INTRODUCTION

Side Channel Analysis (SCA) (*Mangard, Oswald & Popp, 2010*) refers to bypassing the tedious analysis of encryption algorithms, by using the information (such as execution time, power consumption, electromagnetic radiation, *etc.*) leaked by the hardware device embedded in the encryption algorithm during the calculation process, combined with statistical analysis methods to attack cryptographic systems. The side-channel cryptanalysis method is divided into profiling methods and non-profiling methods: non-profiling methods include differential power attack (DPA) (*Kocher, Jaffe & Jun, 1999*), correlation power attack (CPA) (*Brier, Clavier & Olivier, 2004*) and mutual information attack (MIA) (*Gierlichs et al., 2008*); profiling methods include template attack (TA) (*Chari, Rao & Rohatgi, 2002*), side-channel cryptography attack based on multi-layer perceptron (MLPSCA), and side-channel cryptography attack based on convolutional neural networks (CNNSCA). Although the attack method of the non-profiling method is simple and direct, weak side-channel signal or excessive environmental noise can cause the attack to fail. The profiling method can effectively analyze the characteristics of the side-channel signal when the encryption knowledge of the attacking device is obtained in advance, so it is easier to crack the cryptogramme. In the case of an encrypted implementation copy, the best cryptanalysis attack in the traditional SCA method is TA (*Chari, Rao & Rohatgi, 2002*; *Lerman, Bontempi & Markowitch, 2014*; *Picek, Heuser & Guilley, 2017*; *Choudary & Kuhn, 2013*), but TA has difficulties in statistical analysis when processing high-dimensional side-channel signals, and cannot attack the implementation of protected encryption. With the rapid development of supervised machine learning algorithms, it can effectively analyze one-dimensional data with similar power consumption traces in other fields, and side-channel cryptanalysis based on machine learning (MLSCA) (*Lerman et al., 2015*; *Lerman, Bontempi & Markowitch, 2015*; *Picek et al., 2017*) has begun to emerge. The new profiling method MLPSCA surpasses the traditional profiling method in attack performance (*Picek et al., 2017*; *Benadjila et al., 2018*; *Maghrebi, Portigliatti & Prouff, 2016*), and overcomes the shortcomings of template attacks that cannot handle high-dimensional side-channel signals, but it also loses effectiveness when attacking encryption with protection. Nowadays, with the development of machine learning, deep learning techniques with excellent performance in image classification and target recognition have become popular. Studies have shown that the application of convolutional neural network algorithms under deep learning can produce better encryption performance in side-channel analysis (*Benadjila et al., 2018*; *Maghrebi, Portigliatti & Prouff, 2016*; *Picek et al., 2018*; *Cagli, Dumas & Prouff, 2017*; *Dongxin et al., 2018*). The deep network helps to mine the deep features in the data, which can make the neural network have more powerful performance, which makes CNNSCA can also attack the encryption implementation with protection. In the side-channel analysis application scenario, deep learning eliminates the step of manually extracting features from the workflow of model construction. For example, in the traditional bypass attack method, the TA with better attack effect only selects five strong feature points, while the deep learning model can select hundreds to thousands of feature points, select more

features to construct a template, it is extremely beneficial to the generalization and robustness of the side-channel analysis model.

Analyze the above domestic and foreign documents, there are two main types of CNN structures that have successfully used CNNSCA to achieve cryptanalysis, which are based on two variants of Alexnet and VGGnet network structures (*Benadjila et al., 2018*; *Dongxin et al., 2018*; *Dongxin et al., 2019*; *Kim et al., 2019*). Among them, the 2012 ILSVRC (ImageNet Large Scale Visual Recognition Challenge) champion structure Alexnet (*Krizhevsky, Sutskever & Hinton, 2017*), although successful in the SCA application, but in fact, the training accuracy of CNNSCA based on this network variant is not high, moreover, the Alex-CNNSCA network model in the literature (*Dongxin et al., 2018*) has a large amount of training parameters and a long calculation time, which means that there is still room for optimization of this network structure. The 2013 ILSVRC champion network ZFNet (*Zeiler & Fergus, 2014*) has not changed much from the 2012 first ILSVRC champion network Alexnet. The 2014 ILSVRC runner-up structure VGGnet (*Simonyan & Zisserman, 2014*) also succeeded in breaking secrets in the SCA application. In the literature (*Benadjila et al., 2018*; *Dongxin et al., 2019*; *Kim et al., 2019*), VGG-CNNSCA models with different parameters were proposed. Among them, the best cryptanalysis performance is in the literature (*Benadjila et al., 2018*) proposed VGG-CNNSCA, but its training accuracy is still not high. Obviously, there is still room for improvement in the cryptanalysis performance. The 2014 ILSVRC champion network GoogLeNet (*Szegedy et al., 2015*) and the 2015 ILSVRC champion network ResNet (*He et al., 2016*) have also been used in SCA, but the effect is average. This conclusion has been confirmed in the literature (*Benadjila et al., 2018*). The last ILSVRC champion network in 2017 was the SEnet (*Jie et al., 2017*) proposed by Momenta and Oxford University. There is currently little literature on applying this network to SCA scenarios.

Although CNNSCA overcomes the shortcomings of the previous profiling methods and improves the cryptanalysis performance, the existing CNNSCA model learning ability and cryptanalysis performance are not optimal. The disadvantages of these models are: low training accuracy and excessive training time long, taking up too much computing resources, *etc*. The reason is mainly affected by CNNSCA model design and hyperparameter optimization. In order to improve the overall performance of CNNSCA, the paper will improve CNNSCA model design and hyperparameter optimization, and has done the following work:

1. The composition of the CNN architecture in the SCA application scenario is studied, and the calculation process of the CNN core algorithm for side-channel leakage of one-dimensional data is deduced.

2. Taking advantage of the high efficiency of classification and fitting of the VGG-CNNSCA model and the advantages of the Alex-CNNSCA model occupying less computing resources, a new basic model of CNNSCA is designed to better reduce the gradient dispersion of error back propagation in the deep network. The problem is that the SE module is newly embedded in this basic model, so that the model basically

achieves the purpose of breaking the secrets, thereby solving the problem of constructing the CNNSCA model.

3. Apply the above basic model to a known first-order mask data set of the side-channel leak public database (ASCAD). In this application scenario, according to the model design rules and actual experimental results, unnecessary experiments are maximized parameter, optimize the various hyperparameters of the model in the most objective experimental parameter interval to improve the breaking performance of the new CNNSCA, which solves the problem of hyperparameter optimization, and gives the final determination benchmark for hyperparameters. Finally, a new CNNSCA model optimized architecture for attacking unprotected encryption devices-CNNSCAnew is obtained.

4. The performance verified on public data sets exceeds other profiling SCA methods.

The algorithms involved in the paper experiments are all programmed in the Python language, and use the deep learning architecture Keras library (*Eldeeb et al., 2015*) (version 2.4.3) or directly use the GPU version of the Tensorflow library (*Abadi et al., 2015*) (version 2.2.0). The experiment was carried out on an ordinary computer equipped with 16 GB RAM and 8G GPU (Nvidia GF RTX 2060). All experiments use side-channel leaking public data sets-known first-order mask data sets in the ASCAD database, use 50,000 pieces of data from its training set to train the model, and randomly select 1,000 pieces of data from its test set for testing. When testing the cryptanalysis performance of the CNNSCA model, the guessing entropy index is used to evaluate the cryptanalysis performance.

## MATERIALS AND METHODS
### Materials
#### CNN

Convolutional Neural Network (CNN) is one of the most successful algorithms of artificial intelligence, and it is a multi-layer neural network with a new structure. Its design is inspired by the research on the optic nerve receptive field (*Hubel & Wiesel, 1968*; *Lecun & Bengio, 1998*). The core component of CNN, the convolution kernel, is the structural embodiment of the local receptive field. It belongs to the deep network of back propagation training. It uses the two-dimensional spatial relationship of the data to reduce the number of parameters that need to be learned, and improves the training performance of the BP algorithm (Error Back Propagation, which is used to calculate the gradient of the loss function with respect to the parameters of the neural network) to a certain extent. The main difference between CNN and MLP is the addition of the convolution block structure. In the convolution block, a small part of the input data is used as the original input of the network structure, and the data information is forwarded layer by layer in the network, and each layer uses several convolution cores to extract features of the input data. Convolutional neural networks have been successfully applied in computer vision, natural language processing, disaster climate prediction and other fields, especially shine on ILSVRC (*Russakovsky et al., 2015*). ILSVRC is one of the most popular and authoritative

**Table 1 CNN with outstanding performance in previous ILSVRC competitions.**

| Year | Network/Ranking | val top-1 (%) | val top-5 (%) | test top-5 (%) | Remarks |
|------|------------------|---------------|---------------|----------------|---------|
| 2012 | Alexnet (Champion) | 36.7 | 15.4 | 15.32 | 7CNNs, Used data from 2011 |
| 2013 | ZFnet (Champion) | – | – | 13.51 | The result on the ZFNet paper is 14.8 |
| 2014 | VGG (Runner-up) | 23.7 | 6.8 | 6.8 | Post-race, 2 nets |
| 2014 | Googlenet v4 (Champion) | 16.5 | 3.1 | 3.08 | Post-race, v4+Inception-Res-v2 |
| 2015 | Resnet (Champion) | – | – | 3.57 | 6 models |
| 2016 | Trimps-Soushen (Champion) | – | – | 2.99 | Public Security III (additional data) |
| 2017 | SEnet (Champion) | – | – | 2.25 | Momenta and Oxford University |

academic competitions in the field of machine vision in recent years, representing the highest level in the field of imaging. The introduction of outstanding CNNs in the image classification and target positioning projects of the ILSVRC competition over the years is shown in Table 1 (CNN with outstanding performance in previous ILSVRC competitions).

Table 1 sorts out the champion networks and individual runner-up networks of the last ILSVRC classification task from 2012 to 2017, and briefly introduces their names, rankings, classification results under the top1 and top5 indicators, and some remarks. Top1 refers to the largest probability vector as the prediction result, if the classification is correct, it is correct. Top5 is correct as long as there is a correct classification in the top five of the largest probability vectors. Among them, the error rate of the classification results of the last champion network SEnet (2017) under the top5 index is obviously the lowest, reaching 2.25%. Deep convolutional networks have greatly promoted the development of various fields of deep learning.

### CNNSCA model hyperparameters

Hyperparameters of neural network models are a concept often used in machine learning or deep learning, including the structural parameters and training parameters of the network model. To design a CNN model in SCA application scenarios, all the parameters that need to be set are as follows:

(1) Structural parameters
Define all the parameters of the neural network architecture, including the regular parameter network layer activation function, classification function, loss function, and optimizer. In the convolutional neural network, the network layer is subdivided into convolutional blocks (a combination of different numbers of convolutional layers and pooling layers), convolutional layers, full link layers, pooling layers, the number of convolution kernels, convolution kernel size and fill.

The convolution block, convolution layer, pooling layer, number of convolution kernels, convolution kernel size and padding in these parameters mainly control the scale and performance of feature extraction in the feature extraction stage of the CNNSCA model. Full link layer, activation function, classification function and loss function, these

parameters constitute the main body of the CNNSCA network, and perform feature learning and fitting classification on side-channel leakage data.

(2) Training parameters

Control the parameters of the network model training phase, including the number of iterations, batch learning volume, and learning rate. When training a network model, a complete training set is processed at one time, which is called complete batch learning. If a single training sample is processed at a time, it is called random learning. In practice, in order to improve efficiency, a compromise method is usually adopted, called small-batch learning, that is, small batches of training samples are processed at one time during the model learning process. The batch size depends on environmental factors (*Kim, Lee & Nam, 2018*) (such as network architecture, computer GPU performance, the trade-off between network regularization effect and stability, *etc.*). The number of iterations is an important parameter to be adjusted. A small value will cause the network model to underfit (the model is too poor to capture the feature trend in the training data set), while a higher value will cause the network model to overfit (the model is too Complex, perfectly fits the training data set, but cannot generalize its prediction to other data sets). In addition, the variable that optimizes the training effect of the network model-the learning rate (also called the step size), aims to promote the gradient (*i.e.*, the error gradient) drop during the training process.

The number of iterations and the amount of batch learning affect the degree of model training, and the optimizer and learning rate are used to control the gradient of the error. These parameters all have an important impact on CNNSCA's cryptanalysis performance and need to be adjusted according to specific attack scenarios.

## CORE ALGORITHM AND NETWORK STRUCTURE OF CNNSCA

### CNN network structure for SCA

Combined with the side-channel cryptanalysis scenario, the CNN applied to the side-channel attack mainly has six network layers stacked layer by layer and an embeddable SE module:

a) Convolutional layers (Conv for short) are linear layers. The incomplete connection between layers can avoid the two shortcomings of a fully connected network: training weights requires a huge amount of calculation and model overfitting. The weights of the same convolution kernel (also known as filters) in the same layer are shared, allowing the convolution layer to extract constant displacement features while reducing parameters. The convolutional layer can also use multiple convolution kernels. Each convolution kernel extracts different abstract features from the input vector. These abstract features are arranged side by side in an additional dimension (the so-called depth), making the CNN resistant to time-domain distortion Vector features (*Choi et al., 2016*). The convolutional layer usually needs to set the padding mode, one is valid padding, so that the dimension of the feature vector after convolution is smaller than the

original vector; the other is the same padding, so that the convolutional The feature vector dimension is the same as the original vector.

b) Batch Normalization layers (*Ioffe & Szegedy, 2015*) (BN for short), whose role is to reduce the deviation of covariates in the two stages of training and prediction, which is conducive to the use of a higher learning rate for the network model (*Goodfellow, Bengio & Courville, 2016*).

c) Activation layers (ACT for short) are non-linear layers and consist of a single real function, which acts on each coordinate of the input vector. The ReLU function is currently the first choice in deep learning.

d) Pooling layers (POOL for short) are non-linear layers. Use the pooling window to slide on the input vector to extract salient feature points to reduce the feature dimension. There is no weight in the pooling layer, which will not cause distortion of the input signal.

e) Fully-Connected layers (FC for short), the neurons between the layers are completely connected, and these layers need to train a lot of weights. This layer is expressed by an affine function as: D-dimensional x vector is the input, and Ax+B is the output. Among them, A∈RC×D is the weight matrix and B∈RC is the deviation vector. These weights and deviations are the training parameters of the FC layer.

f) Softmax layer (SOFT for short). In multi-classification tasks, softmax is usually used as the activation function of the output layer. Here, softmax is used to represent the output layer. This layer classifies the input, obtains the predicted value of each label, and takes the label corresponding to the maximum value as the global classification result.

g) SE module, SEnet is a classic attention model structure, and it is also a required basic network structure for fine-grained classification tasks. SEnet proposed the Squeeze-and-Excitation (SE) module, which did not introduce a new spatial dimension, and improved the representation ability of the model by displaying the channel correlation between the features of the convolutional layer. The feature recalibration mechanism: by using global information to selectively enhance informatized features and compress those useless features at the same time. In deep network training, this mechanism can effectively overcome the gradient dispersion problem in error back propagation. The SE module is universal. Even if it is embedded in an existing model, its parameters do not increase significantly. It is a relatively successful attention module (*Jie et al., 2017*). The structure of the SE module is shown in Fig. 1 (SE module).

In Fig. 1, the SE module uses global pooling as a squeeze operation, and then uses two FC layers to form an excitation structure to profile the correlation between channels, and output and input the same number of feature channels weights. The advantages of this are: (1) it has more nonlinearity and can better fit the complex correlation between channels; (2) the amount of parameters and the amount of calculation are greatly reduced. Then obtain the normalized weight between 0 and 1 through a sigmoid function, and then use a scale operation to weight the normalized weight to the features of each channel

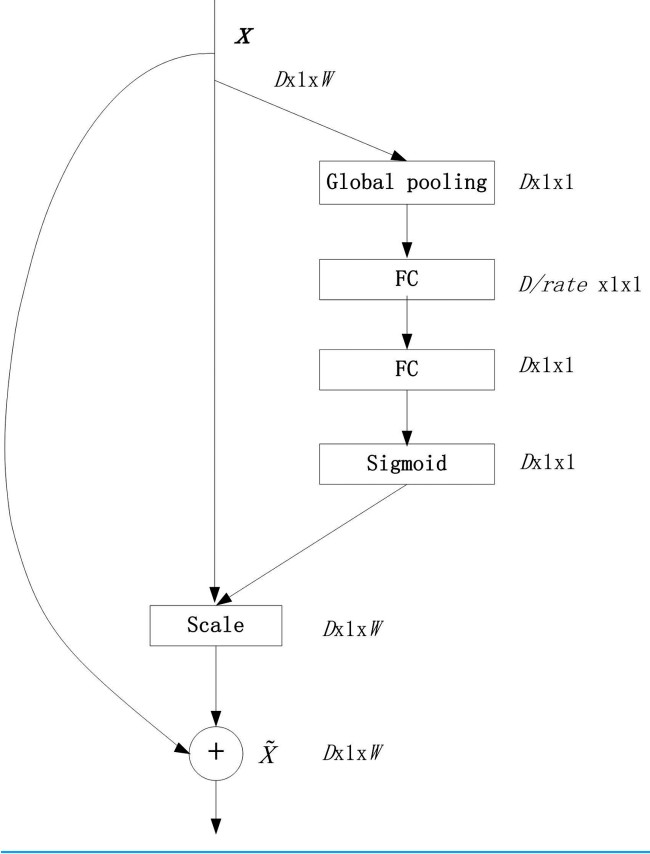

**Figure 1 SEnet module.**

(*Jie et al., 2017*). Finally, the output of scale is superimposed on the input $x$ before the SE module to generate a new vector $\tilde{x}$.

## Core algorithm of CNN for SCA

(1) Convolution calculation

Usually convolution operations in the field of computer vision are numerical operations on two-dimensional image data. In the SCA application scenario, the dimensionality of the convolution operation is adjusted, which is to slide the convolution kernel on the one-dimensional energy trace data. The number of steps moved each time is called the step length, and the convolution calculation is performed on each sliding to obtain a value. After one round of calculation is completed, a feature vector representing the vector feature is obtained. The rule of numerical operation is to multiply a one-dimensional convolution kernel with a value at the corresponding position of a one-dimensional vector, and then sum. For example, there is a $1 \times 3$ convolution kernel, which convolves a $1 \times 6$ one-dimensional vector with a step size of 1. The calculation process is shown in Fig. 2 (Convolution calculation process).

In Fig. 2A, the convolution kernel slides from the left side of the input vector. The first numerical calculation is: $1 \times 1 + 0 \times 0 + 1 \times 1 = 2$, and the first value 2 of the new feature

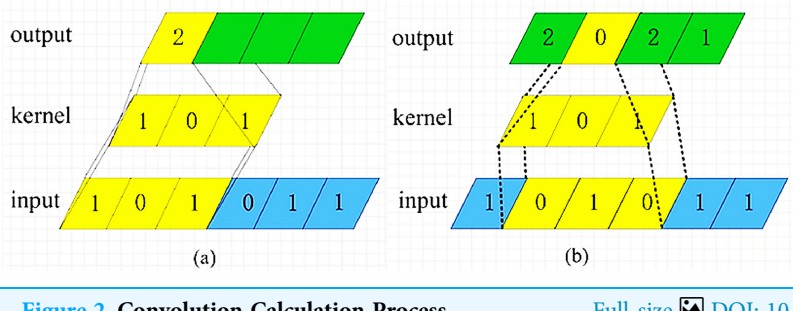

**Figure 2  Convolution Calculation Process.**               

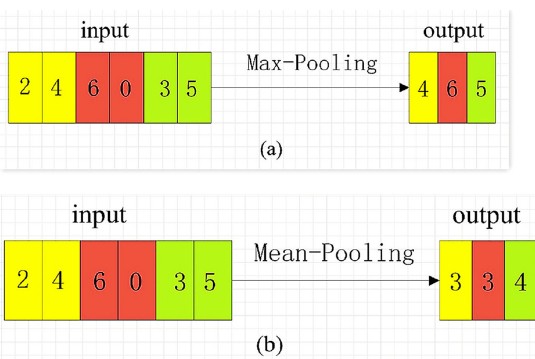

**Figure 3  Pooling Calculation Process.**               

vector is obtained. Then, the convolution kernel slides one step to the right to continue the numerical calculation: $1 \times 0 + 0 \times 1 + 1 \times 0 = 0$, to get the second value 0 of the new feature vector, as shown in Fig. 2B. Repeat this process until the convolution kernel slides to the far right of the input vector, and the convolution calculation is complete.

(2) Pooling calculation

   There are three ways of pooling: Max-Pooling, Mean-Pooling and Stochastic Pooling. Maximum pooling is to extract the maximum value of the value in the pooling window, average pooling is to extract the average value of the value in the pooling window, and random pooling is to randomly extract the value in the pooling window. The original pooling operation of CNN is also a numerical operation on two-dimensional image data. In the SCA application scenario, the pooling calculation has also been dimensionally adjusted, and a pooling mode is selected for calculation on the one-dimensional energy trace data. For example, the pooling window size is $1 \times 2$, and the maximum or average pooling operation is performed on a $1 \times 6$ one-dimensional vector with a step size of 2. The pooling calculation is shown in Fig. 3 (Pooling calculation process).

   In Fig. 3A, the maximum pooling starts from the left side of the input vector. Every two steps of the pooling window, the maximum value of the two values in the window is selected as a value of the new feature vector. The average pooling is shown in Fig. 3B. For every two sliding steps of the pooling window, the average of the two values of the window

class is calculated as a value of the new feature vector. The pooling window slides to the right until the rightmost of the input vector, and the pooling calculation is complete.

(3) softmax function

This function normalizes the output value and converts all output values into probabilities. The sum of the probabilities is 1. The formula of softmax is:

$$softmax(x_i) = \frac{exp(x_i)}{\sum_j exp(x_j)} \tag{1}$$

Here $x_i$ represents the input of the i-th neuron in the softmax layer, $x_j$ represents the input of the j-th neurons in the softmax layer, and $\sum_j$ is the sum of calculations for $x_j$. The result of the function is used as the fitting probability of the i-th neuron label.

(4) Principle of weight adjustment

Using the cost function and gradient descent algorithm (*Liqun & Qian, 2005*), each time the network model is trained, the weights are automatically adjusted in the direction of error reduction, so that the training parameters are repeated until all iterations are over, and the weight adjustment is completed.

(5) Evaluation of Cryptanalysis Performance

Generally, security officers consider two indicators when evaluating CNNSCA's cryptanalysis performance: one is the training accuracy of the neural network model during modeling, the Acc indicator (*Hawkins, 2004*), and the other is the security indicator guessing entropy of the key obtained in the attack phase (*Standaert, Malkin & Yung, 2009*; *Masure, Dumas & Prouff, 2020*). The guessing entropy index is commonly used to evaluate the SCA cryptanalysis performance, and the guessing entropy is used to measure the efficiency of decrypt. Guessing Entropy (GE) is obtained through a custom rank function $Rank(\cdot)$, which is defined as:

$$Rank(\hat{g}, D_{train}, D_{test}, n) = |\{k \in K | d_n[k] \geq d_n[k^*]\}| \tag{2}$$

The adversary uses the modeling data set $D_{train}$ to establish a bypass analysis model $g$, and uses $n$ energy trace samples in the attack data set $D_{test}$ to perform $n$ attacks during the attack phase. After each attack, the logarithm value of the distribution probability of 256 types of hypothetical cryptograms is obtained, compose a vector $d_i = [d_i[1],$ $d_i[2], L, d_i[k]]$, whose indexes are arranged in the positive order of the hypothetical cryptogramme's key space (the index counts from zero), where $i \in n$, $k \in K$, and $K$ is the key space of the hypothetical cryptogramme. The results of each attack are accumulated. Then, the rank function $Rank(\cdot)$ sorts all the elements of the vector $d_i$ in reverse order by value, and keeps the position of the corresponding index of each element in the vector before and after sorting consistent with the position of the element, and obtains a new ranking vector $D_i = [D_i[1], D_i[2], L, D_i[k]]$, where each the element $D_i[k]$ contains two

values $k$ and $d[k]$, and finally the index of the logarithmic element of the known cryptogramme $k^*$ probability in $D_i$ is output, that is, the guessing entropy GE($d[k^*]$). At the i-th attack, the higher the matching rate of the energy trace model of the real cryptogramme, the higher the index ranking of its GE($d[k^*]$). Guessing entropy is the GE($d[k^*]$) index ranking output of each attack—*rank*. In $n$ attacks, the better the performance of the cryptanalysis method and the higher the efficiency, the faster the ranking of GE($d[k^*]$) converge to zero. It shows that in the i-th attack, the guessing entropy converges to zero and continues to converge in subsequent attacks. The adversary only needs $i$ attacks to crack the cryptogramme, that is, only $i$ power consumption traces are needed to break the secret. Eq. (2) can be rewritten as (3):

$$GE_n(\hat{g}) = Every[Rank(\hat{g}, D_{train}, D_{test}, n)] \tag{3}$$

(4) Side-channel leaking public data sets

The newly published ASCAD database (*Benadjila et al., 2018*) aims to achieve AES-128 with first-order mask protection, namely 8-bit AVR microcontroller (ATmega8515), in which the energy trace is the data signal converted by the collected electromagnetic radiation. The adversary outputs the collected signal for the third S-box of the first round of AES encryption, and launches an attack against the first AES key byte. The database follows the MNIST database (*Lecun & Cortes, 2010*) rules and provides a total of four data sets, each with 60,000 entries power consumption traces, of which 50,000 power consumption traces are used for analysis/training, and 10,000 power consumption traces are used for testing/attack. The first three ASCAD data sets respectively represent the encryption realization leakage with three different random delay protection countermeasures. The signal offsets desync=0, desync=50, and desync=100 are used to represent these three data sets with two strategies of mask and delay. All power consumption traces in the first three types of data sets contain 700 feature points. These feature points are selected from the original energy trace containing 100,000 feature points, and the selection basis is the position of the largest signal peak. When the mask is known, the maximum signal-to-noise ratio of the data set can reach 0.8, but it is almost 0 when the mask is unknown. The last ASCAD data set stores the original energy trace.

## METHODS

1 Design of CNNSCA base model

With reference to the advantages of the VGG-CNNSCA model with high classification and fitting efficiency and the Alex-CNNSCA model occupying less computing resources, the paper selects the same structural parameters from these two models, and some of the factors that promote the high fitting efficiency of the two types of models parameter. These parameters construct a new CNN simple model specifically for SCA scenarios, which is used to test the impact of different hyperparameters on model performance. The convolution block of this simple model consists of the Conv layer, BN layer, and ACT layer. After the block, a POOL layer is usually added to reduce the feature dimension.

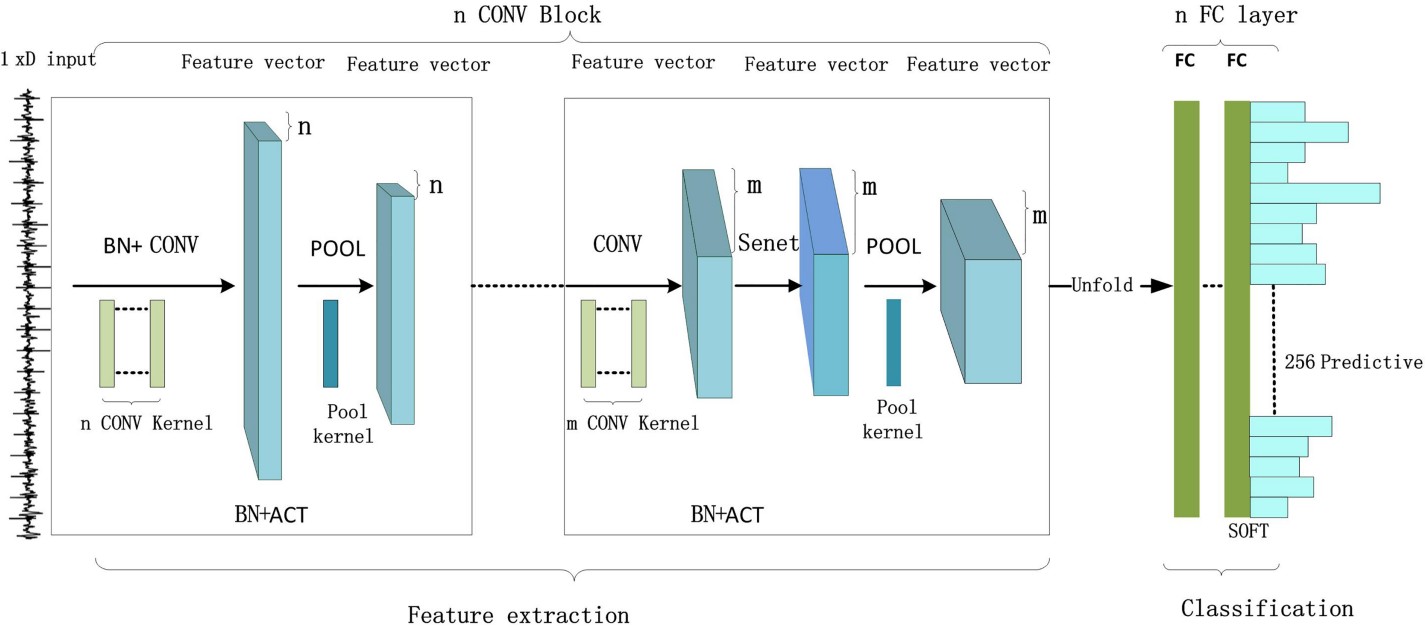

**Figure 4 Convolutional network structure in a side-channel attack scenario.**

The new convolution block is repeated n times in the network model until it is reasonable. Until the output of the size. Then, introduce n FC layers, use the softmax function in the last FC layer, and finally output the classification prediction results. In addition, in order to improve the classification and recognition performance of the CNNSCA model, the SE module is newly embedded in the simple model. Its main function is to reduce the gradient dispersion problem in error back propagation. This is the first use in the CNNSCA model. The SE module will be embedded between the convolutional layer and the pooling layer of the convolutional block of the simple model, and the simple model containing the SE module will be renamed to the CNNSCA base model-CNNSCAbase. The newly designed CNNSCAbase structure is shown in Fig. 4 (Convolutional network structure in a side-channel attack scenario).

The initial configuration basis and selection of CNNSCAbase are as follows: Find out the two prototypes of Alex-CNNSCA and VGG-CNNSCA to set the same parameters, set these parameters in CNNSCAbase in the same way, these parameters are as follows: 5 convolutional blocks, 3 full connections, the padding modes of the convolutional layers are SAME, and the activation functions of all layers before the last layer of the network select ReLU. In addition, in most classification tasks, convolutional networks use softmax, crossentropy, and RMSprop as the model's classification function, loss function, and optimizer (*Krizhevsky, Sutskever & Hinton, 2017*; *Zeiler & Fergus, 2014*; *Simonyan & Zisserman, 2014*; *Szegedy et al., 2015*; *He et al., 2016*). Here, CNNSCAbase also chooses to use these three activation functions. Since the side-channel leakage data belongs to one-dimensional data, the processing complexity is less than that of two-dimensional data. Here, the convolution layer of each convolution block is initialized to 1, and the number of convolution kernels in the first convolution layer is 64 (choose the smaller number of

| Table 2 CNNSCAbase configuration. | | | |
|---|---|---|---|
| **ConvNet configuration** | | | |
| Input(1x700 vector) | | | |
| Block1 | (Conv3-64)x1 | Same\ReLU | AveragePool (2,2) |
| Block2 | (Conv3-128)x1 SE | Same\ReLU | AveragePool (2,2) |
| Block3 | (Conv3-256)x1 SE | Same\ReLU | AveragePool (2,2) |
| Block4 | (Conv3-512)x1 SE | Same\ReLU | AveragePool (2,2) |
| Block5 | (Conv3-1024)x1 SE | Same\ReLU | AveragePool (2,2) |
| (FC-4096)x2, ReLU | | | |
| (FC-256)x1, Soft-max | | | |
| Model compile (crossentropy, RMSprop) | | | |

**Figure 5** Training Effect of CNNSCAnoSE Model and CNNSCAbase model.

Alexnet or VGGnet), the size of the convolution kernel is $3 \times 3$, the step size is 1, the pooling mode is tentatively averaged pooling mode, the pooling window size is 2, and the step size is 2. In addition, in the initial setting of CNNSCAbase, a new SE module is embedded in the last four convolution blocks. All initial structure parameters of CNNSCAbase are shown in Table 2 (CNNSCAbase Configuration).

Here we first verify and analyze the model training effect of CNNSCAbase with and without SE module. Remove the SE module from the CNNSCAbase model, all other parameters remain unchanged, and name this model CNNSCAnoSE. Train the models CNNSCAnoSE and CNNSCAbase on the training set of the ASCAD dataset with known masks. The training results of the two models are shown in Fig. 5 (Training effect of CNNSCAnoSE model and CNNSCAbase model).

As shown in Fig. 5, when the training iteration reaches 28 times, the accuracy of CNNSCAbase is significantly higher than that of CNNSCAnoSE, which is about 96%. Continue to train the CNNSCAnoSE model, and when the training iteration reaches 70 times, its accuracy rate rises to about 90%. In addition, when the training accuracy of the two models is close, the training time of the 28-iteration CNNSCAbase model is about 1,393 s, which is significantly less than the training time of the 70-iteration CNNSCAnoSE model, the training time of the latter is about 2,240 s. This proves that the SE module can promote the improvement of the classification performance of the CNNSCAbase model and can reduce the model training time.

Next, we discuss the hyperparameter optimization of the CNNSCA model. Model structure parameters and training parameters are hyperparameters and need to be set in advance. Later, we will design a set of experimental procedures to optimize these hyperparameters in specific application scenarios. For example, we choose to determine the model parameters first rather than the global training parameters, first determine the number of Conv layers, rather than first determine the kernel size or the number of filters. The reason for this design is: currently, Python's deep learning architecture library (*Eldeeb et al., 2015*; *Abadi et al., 2015*) is mainly used to program the CNN network. When using these library methods, the CNN network structure is usually programmed first. The order in which these parameters appear in the program code will be affected by the library methods, and then the training parameters are designed according to the size of the network and the size of the training set. It is precisely in consideration of the order in which the parameters appear during programming, we have designed the order of the following experimental procedures.

2 Selection and optimization of CNN structure parameters for side-channel cryptanalysis

2.1 Structural parameter selection rules

In section Methods 1, the base model CNNSCAbase is set, and the best model after parameter optimization will be named CNNSCAnew later. In CNNSCAbase, in addition to the specific set of structural parameters, the remaining structural parameters need to be customized. These structural parameters include classification function, loss function, optimizer, the number of convolutional layers in each convolution block, the number of convolution kernels in the convolution layer, convolution kernel size, pooling layer pooling mode. When choosing these custom structure parameters, you need to follow the classic rules of building a deep learning network structure (*Zeiler & Fergus, 2014*; *Szegedy et al., 2015*), which can reduce the number of unnecessary test parameters. The rules are as follows:

Rule 1: Set the same parameters for the convolutional layers in the same convolutional block to keep the amount of data generated by different layers unchanged.

Rule 2: The dimensionality of each pooling window is 2, and the window sliding step is also 2, each operation reduces the dimensionality of the input data to half.

Rule 3: In the convolutional layer of the i-th block (starting from i = 1), the number of convolution kernels is n: $n_i = n_1 \times 2^{i-1}$, i ≥ 2. This rule keeps the amount of data processed by different convolution blocks as constant as possible. The network structure characteristics of VGG-16 in this reference (*Simonyan & Zisserman, 2014*) are formulated.

Rule 4: The size of the convolution kernel of all convolution layers is the same.

2.2 Structural parameter optimization

Among the custom structure parameters, the structure parameters that need to be further adjusted through experimental analysis are: the number of convolution layers in each convolution block, the number of convolution kernels in the convolution layer, the

**Table 3 CNNSCAbase.Conv1-7 configuration.**

**Conv configuration**

| Block | Conv1 | Conv2 | Conv3 | Conv4 | Conv5 | Conv6 | Conv7 |
|---|---|---|---|---|---|---|---|
| Block1 | (Conv3-64)x1 | (Conv3-64)x2 | (Conv3-64)x2 | (Conv3-64)x1 | (Conv3-64)x1 | (Conv3-64)x1 | (Conv3-64)x1 |
| Block2 | (Conv3-128)x1 SE | (Conv3-128)x2 SE | (Conv3-128)x2 SE | (Conv3-128)x2 SE | (Conv3-128)x1 SE | (Conv3-128)x1 SE | (Conv3-128)x1 SE |
| Block3 | (Conv3-256)x1 SE | (Conv3-256)x2 SE | (Conv3-256)x2 SE | (Conv3-256)x2 SE | (Conv3-256)x2 SE | (Conv3-256)x1 SE | (Conv3-256)x1 SE |
| Block4 | (Conv3-512)x1 SE | (Conv3-512)x2 SE | (Conv3-512)x2 SE | (Conv3-512)x2 SE | (Conv3-512)x2 SE | (Conv3-512)x2 SE | (Conv3-512)x1 SE |
| Block5 | (Conv3-1024)x1 SE | (Conv3-1024)x2 SE | (Conv3-1024)x3 SE | (Conv3-1024)x2 SE | (Conv3-1024)x2 SE | (Conv3-1024)x2 SE | (Conv3-1024)x2 SE |

size of the convolution kernel, the pooling mode of the pooling layer, SE module. The experimental process of structural parameter optimization is as follows:

(1) Number of convolutional layers

In Section Methods 1, in the initial CNNSCAbase structure, the number of convolutional layers for each convolution block is 1, and the convolutional structure is named Cnov1. Refer to the number of convolutional layers of different convolutional blocks of the Alexnet and VGGnet16 prototypes. It is found that the minimum number is 1 and the maximum is 3, and the small number is distributed in the front convolution block, and the large number is at the back. This is also to build deep learning The common habit of the Internet. Therefore, the upper limit of the number of convolutional layers of the CNNSCAbase convolution block is set to 3, and the baseline is Cnov1, and a certain convolutional layer parameter configuration can be obtained through two sets of necessary experiments. When training the CNNSCAbase model, the training iteration and batch parameters of the current optimal CNNSCA model (*Benadjila et al., 2018*) are used, which are 75 and 200 (in all experiments in section Methods 2.2, unless otherwise specified, the iteration and batch parameters are used. experiment).

Experiment 1: Set up a model in which the number of convolutional layers in 5 convolutional blocks is 2, and other parameters are consistent with CNNSCAbase, and the structure is named Cnov2. Then set the number of convolutional layers of the first 4 convolutional blocks to 2, and the convolutional layer of the last convolutional block to 3. Other parameters are consistent with CNNSCAbase, and the structure is named Cnov3. The specific settings of the number of convolutional layers of each convolution block of Cnov1~3 are shown in Table 3 (CNNSCAbase.Conv1–7 Configuration). The three structures constructed are trained and tested, and the results of experiment 1 are shown in Fig. 6 (Convergence of guessing entropy of Cnov1~3).

From the results in Fig. 6, it is found that when Cnov2 and Cnov1 attack the 750th energy trace, their guessing entropy basically converges to 0, while Cnov3 cannot converge in a finite number of (1,000) attacks. When doing further analysis, if you set two or more convolutional blocks with 3 convolutional layers in the 5 convolutional blocks of the

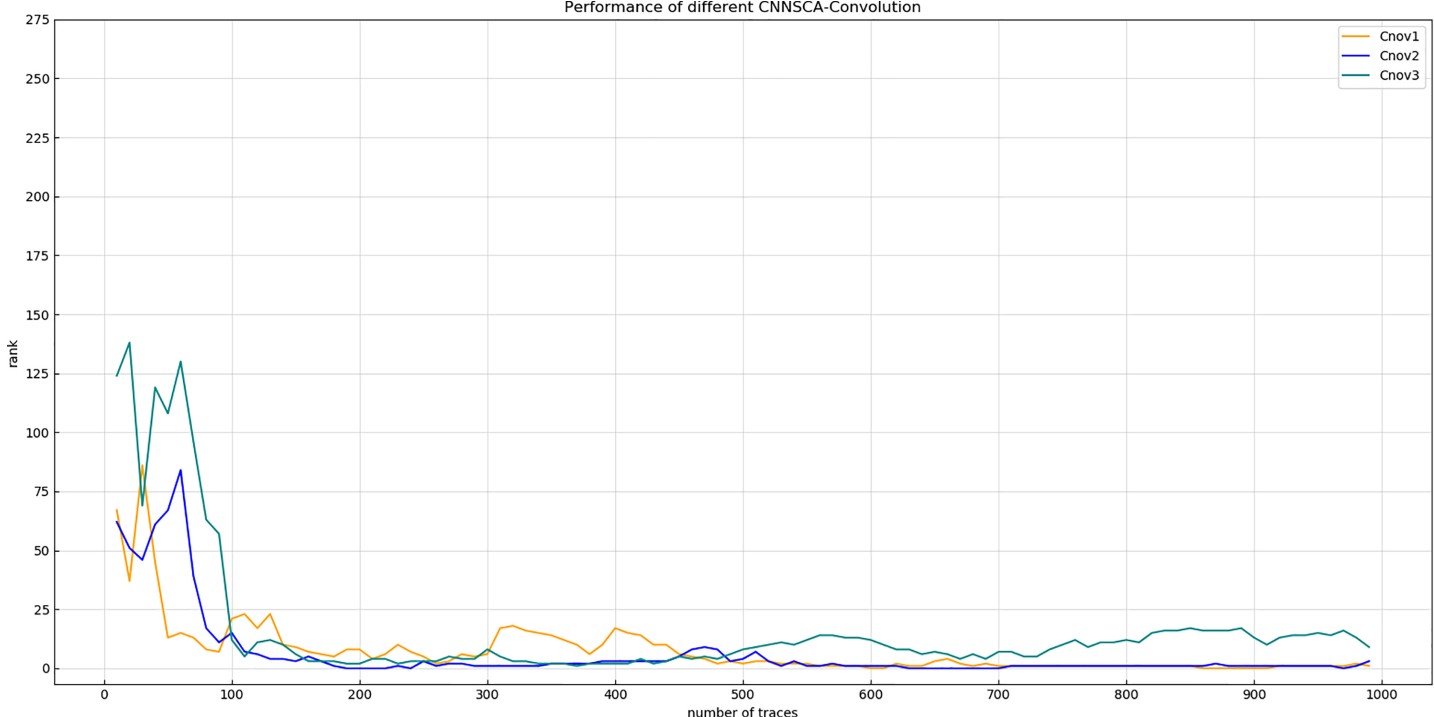

**Figure 6 Convergence of guessing entropy of Cnov1~3.** Each curve represents the convergence trend of guessing entropy under the six model structures of Cnov1~3. The abscissa represents the number of energy traces used in the attack, and the ordinate represents the ranking of the guessing entropy.                                

model, the calculation amount and parameter amount of model training will increase by several times, and the 8G GPU memory used in the experiment will be directly exhausted, unable to run the code, then this parameter setting method will have no practical significance. Therefore, the upper limit of the number of convolutional layers for each convolutional block is determined to be 2.

Experiment 2: On the basis of the conclusion of Experiment 1, the convolutional layer parameter setting of each convolution block is further accurate. As shown in Fig. 6, the convergence of the orange line representing the entropy of Cnov2's guess is more stable than that of Cnov1, but it is obvious that there are more convolutional layers, which means that the amount of model calculations and parameters are relatively large, which affects the overall performance of the model. Therefore, four structures of Cnov4~7 are set, and each structure sequentially reduces the number of convolutional layers in each convolution block of Cnov2 by one. The specific settings of the number of convolutional layers of each convolution block of Cnov4~7 are shown in Table 3 (CNNSCAbase. Conv1–7 Configuration). Train and test these constructed structures, and the results of experiment 2 are shown in Fig. 7 (Convergence of guessing entropy of Cnov1~2,4~7).

The red curve representing the entropy of Cnov5 guessing in Fig. 7 converges optimally. Finally, the number of convolutional layers of the Cnov5 structure is determined, and the benchmark is set for the number of convolutional layers of CNNSCAnew.

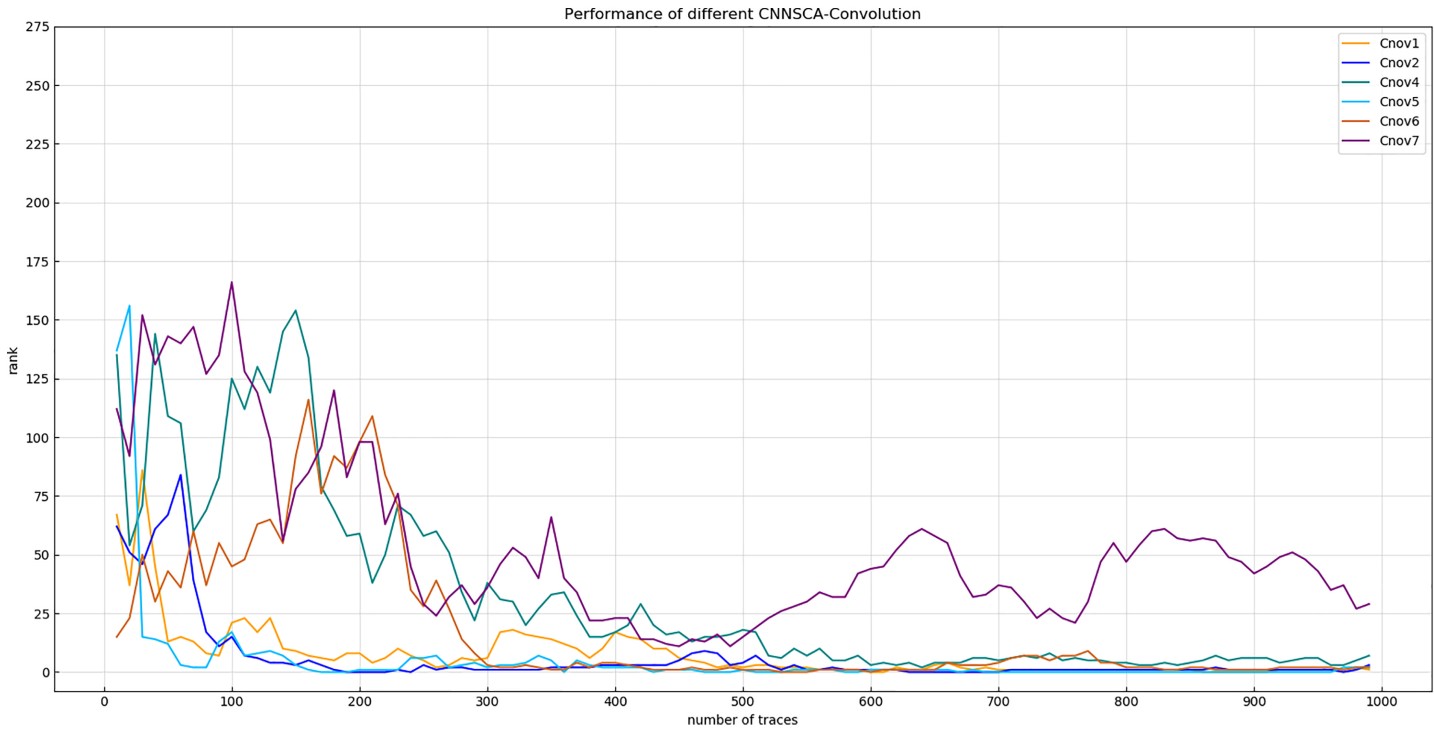

**Figure 7 Convergence of guessing entropy of Cnov1~2,4~7.** Each curve represents the convergence trend of guessing entropy under the six model structures of Cnov1~2,4~7. The abscissa represents the number of energy traces used in the attack, and the ordinate represents the ranking of the guessing entropy.

1. The number of convolution kernels in the convolution layer

It is known that the number of convolution kernels of each convolutional layer of CNNSCAbase is initially set according to Rule 3. The number of convolution kernels of the first convolutional layer is 64. Usually increasing the number of convolution kernels means that more dimensional feature extraction is performed on the input data, thereby improving the classification efficiency of the convolutional network. But it will inevitably lead to an increase in the amount of calculation and storage of the attack device, which will lead to an increase in the training time of the model. Therefore, under the condition that the efficiency loss of the guarantee model is not large, the model training time can be reduced by reducing the number of convolution kernels. Since the number of convolution kernels in the later layer increases by a factor of 2 of the number of convolution kernels in the first convolution layer, to determine the number of convolution kernels as a benchmark, it is only necessary to test the number of convolution kernels in the first convolution layer. At the same time, the CNNSCA model in reference (*Benadjila et al., 2018*), the upper limit of the number of convolution kernels in the convolution layer is 512, which can achieve the effect of breaking the density, so the paper also adjusts the upper limit of the number of convolution kernels to 512.

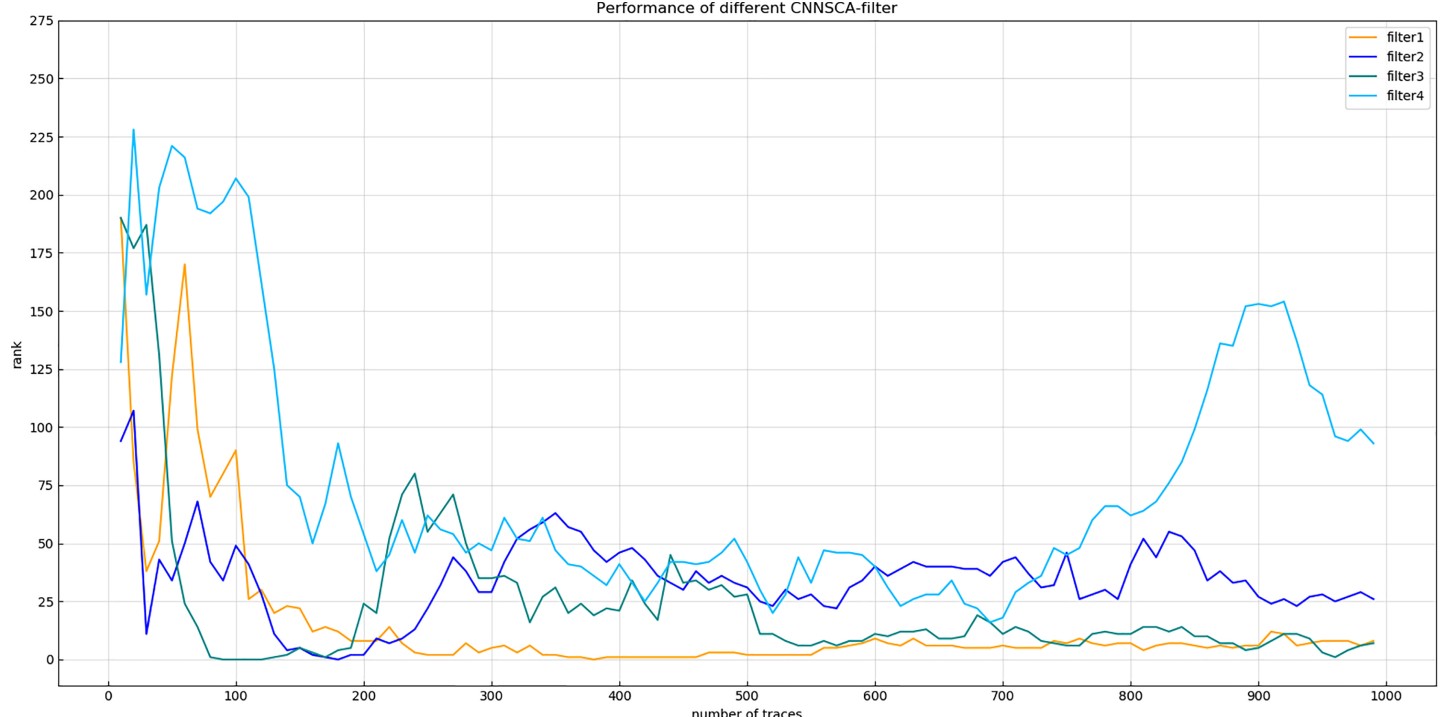

**Figure 8 Convergence of guessing entropy of filter1~4 (epochs=75).** Each curve represents the model guessing entropy of the four convolution kernel sizes. The abscissa represents the number of energy trajectories used in the attack, and the ordinate represents the order of guessing entropy.

Experiment 3: Name the four structures tested as filter1, filter2, filter3, and filter4. The convolution kernel values of the first convolution layer are 8, 16, 32, 64, and the number of convolution kernels of the remaining four convolution blocks is also increased by a factor of 2 respectively. The upper limit of the number of convolution kernels is always 512. Other structural parameters are the parameters of the current CNNSCAnew. Train and test the filter1~4 structure, and the result of experiment 3 is shown in Fig. 8 (Convergence of guessing entropy of filter1~4 (epochs=75)).

Figure 8 shows that after 75 iterations of training, the guessing entropy of the filter4 structure cannot converge. Although the guessing entropy of the filter1~3 structure converges, it fluctuates all the time. When checking the training accuracy of the filter1~3 structure, it is found that the accuracy of the three structures has reached more than 99%, or even reached 1. Obviously, the model has an overfitting phenomenon, which is the most common problem in neural networks. Therefore, the number of training iterations of the filter1~3 structure is reduced to 40, the three structures are retrained, and then the test set is attacked again, and the result shown in Fig. 9 (Convergence of guessing entropy of filter1~3 (epochs=40)) is obtained. The guessing entropy of the filter1~3 structure in Fig. 9 all converge to rank 0, and filter3 converges to the position of rank 0 earliest. In summary, the benchmark for selecting convolution kernel parameters is the filter3 structure, and the convolution kernel parameters of the CNNSCAnew structure are updated synchronously.

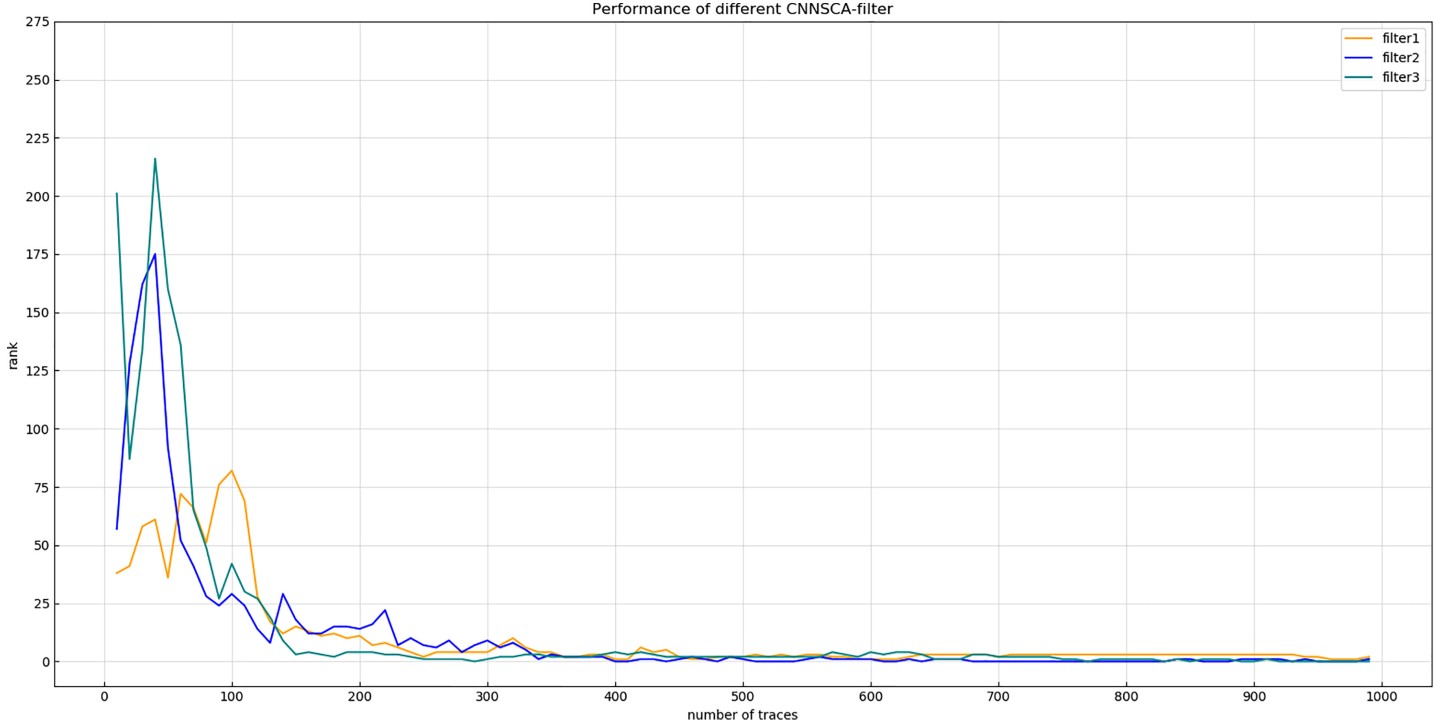

**Figure 9 Convergence of guessing entropy of filter1~3 (epochs=40).** Each curve represents the model guessing entropy of the three convolution kernel sizes. The abscissa represents the number of energy trajectories used in the attack, and the ordinate represents the order of guessing entropy.

(3) Pooling mode of pooling layer

It is known that the initial setting mode of the pooling layer of CNNSCAbase is AveragePool, and another common pooling mode is MaxPooling. According to rule 3, both the pooling window and the pooling step size are still selected here.

Experiment 4: Will test the impact of two pooling modes AveragePool and MaxPooling on the current CNNSCAnew structure. The result of experiment 4 is shown in Fig. 10 (Convergence of guessing entropy of AveragePool and MaxPool structure).

In Fig. 10, it is obvious that the guessing entropy convergence of the average pooling structure is better than the maximum pooling structure, so the benchmark of the pooling layer pooling mode is average pooling, and the pooling mode of the CNNSCAnew structure is set to average pooling.

(4) Convolution kernel size

The size of the convolution kernel of each convolution layer in CNNSCAbase is initially set to $1 \times 3$, or 3 for short. In deep learning, people often reduce the size of the convolution kernel by increasing the depth of the network, thereby reducing the computational complexity of the network. In the VGG-CNNSCA structure, the convolution kernel uses a larger size of 11, and in the Alex-CNNSCA structure, a small size of 3 is used.

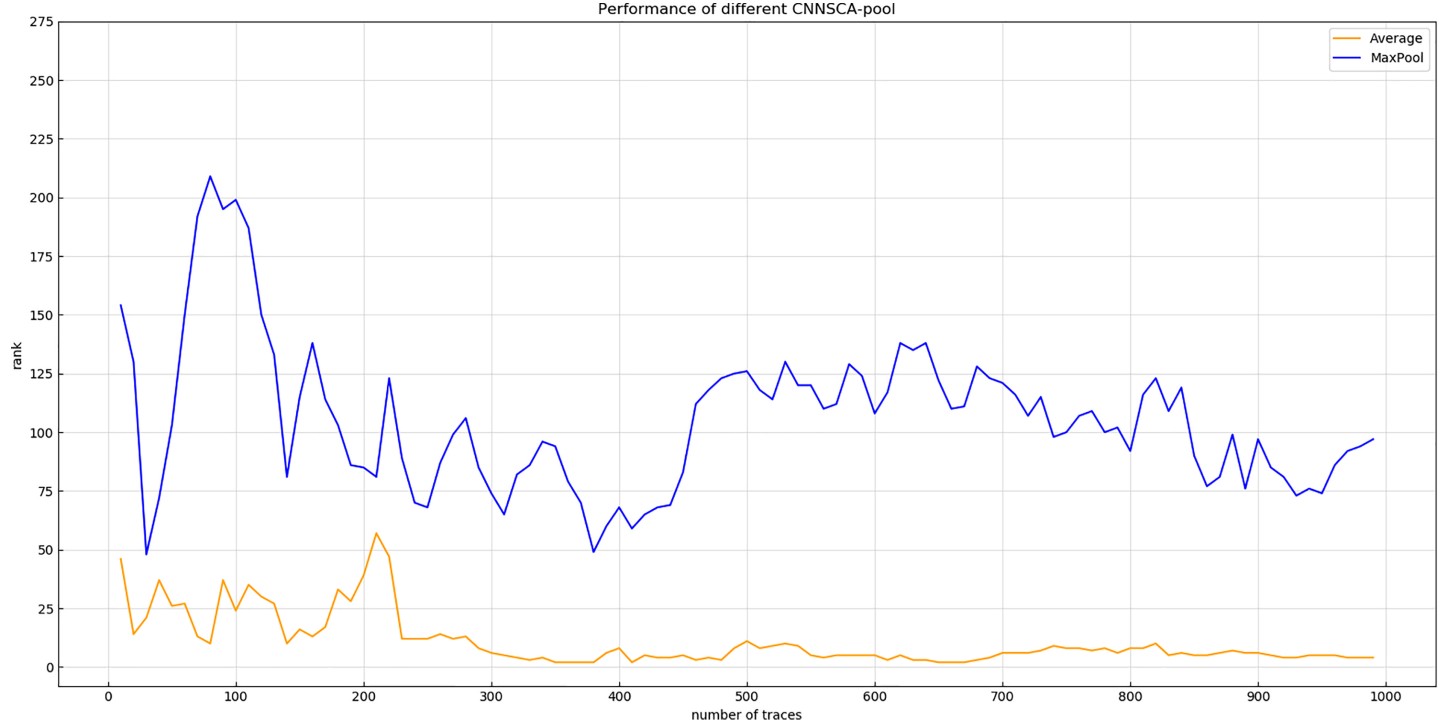

**Figure 10 Convergence of guessing entropy of average pool and MaxPool structure.** Each curve represents the model guessing entropy of two pooling methods. The abscissa represents the number of energy trajectories used in the attack, and the ordinate represents the order of guessing entropy.

Experiment 5: Test the attack effects of the models with the convolution kernel sizes of 3, 5, 7, 9, and 11, and name these five structures as kernel3, kernel5, kernel7, kernel9, and kernel11. The other parameters of these structures are compared with The current CNNSCAnew is the same. The result of experiment 5 is shown in Fig. 11 (Convergence of guessing entropy of different convolution kernel size structures).

In Fig. 11, the convolution kernel size of the structure kernel3 is 3, which guesses that the entropy convergence is better than other structures, so the size 3 is used as the setting reference for the convolution kernel size. At the same time, the size of the convolution kernel of the CNNSCAnew structure is set to 3.

(5) SE module

The attention mechanism in deep learning is essentially similar to the selective visual attention mechanism of humans, and the core role is to select information that is more critical to the current task goal from a large number of information (*Jie et al., 2017*). The paper has initially added an SE fixed module to the last four convolution blocks of CNNSCAbase. The initial setting of the dimensional change ratio of the first full link layer of the SE module is 1/16, but this conventional setting is in the SCA scene The suitability of the medium requires further verification.

Experiment 6: Test the SE module of the model. The rate of the dimensional change of the first full link layer is 1/4, 1/8, 1/16, and 1/32 respectively. The other parameters of the

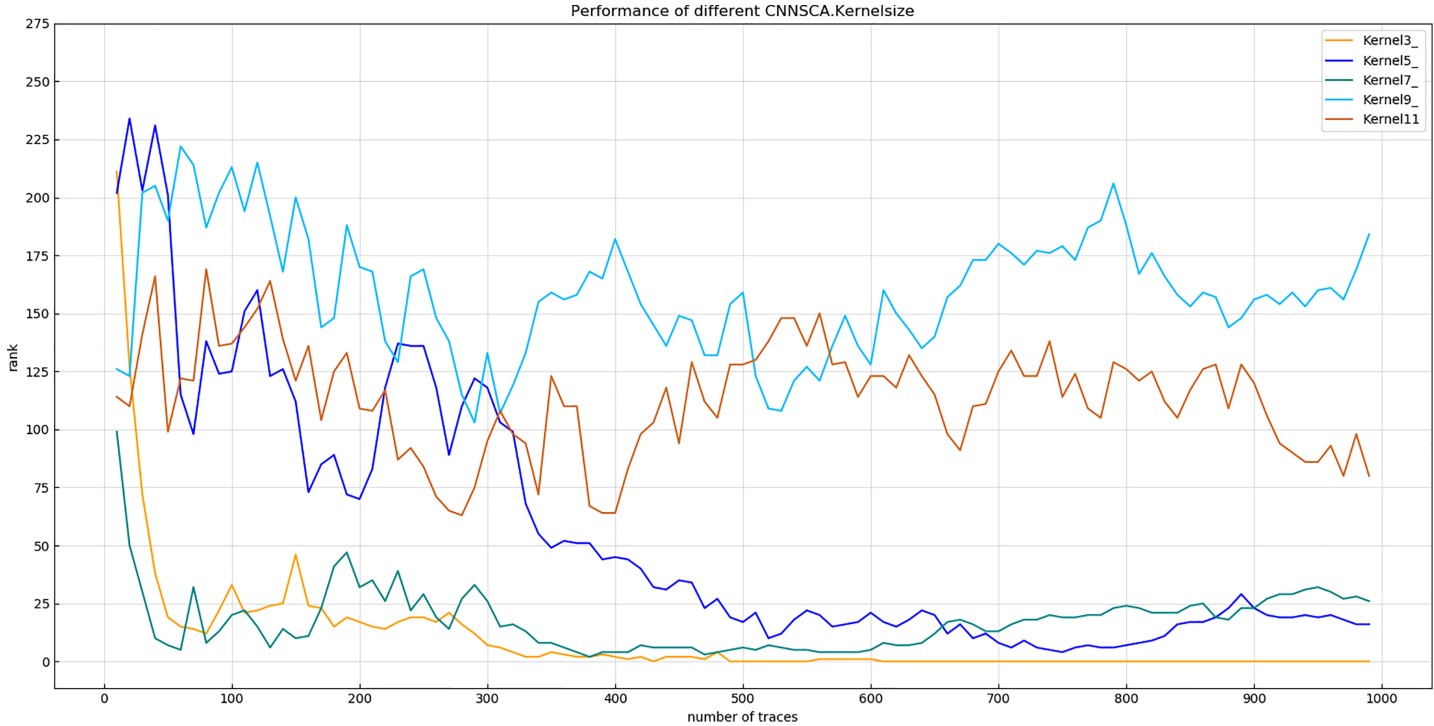

**Figure 11 Convergence of guessing entropy of different convolution kernel size structures.** Each curve represents the model guessing entropy of five convolution kernel sizes structures. The abscissa represents the number of energy trajectories used in the attack, and the ordinate represents the order of guessing entropy.

test model are the same as CNNSCAnew. Experiment 7: Test the attack effect of the SE module used 1, 2, and 3 times for the last four convolutional blocks in the current CNNSCAnew structure. The results of experiment 6 are shown in Fig. 12 (Convergence of guessing entropy for different SE dimension ratios).

As shown in Fig. 12, when the dimensional ratio of the SE module is 1/8, the guessing entropy convergence of the overall structure of the CNN is the best. On the basis of this dimensional ratio, the result of Experiment 7 is shown in Fig. 13 (Convergence of guessing entropy for different number of SE cycles). It is found that when the last four convolution blocks of CNNSCAnew use the SE module twice, the guessing entropy converges fastest. Therefore, the dimensional ratio of the SE module is 1/8 and the SE module is looped twice as a new benchmark for the parameters of the SE module in the CNNSCAnew structure.

(6) Number of channels at the full link layer

The CNNSCA model in literature (*Benadjila et al., 2018*; *Dongxin et al., 2018*) uses 4,096 channels in the fully connected layer, which is similar to the number of channels in the fully connected layer of the original VGGnet and Alexnet structures. Considering that the classification task of the ImageNet competition is 1,000 classifications, and only 256 classifications are needed in the SCA scene, the number of channels can be adjusted appropriately to reduce the training complexity of the model.

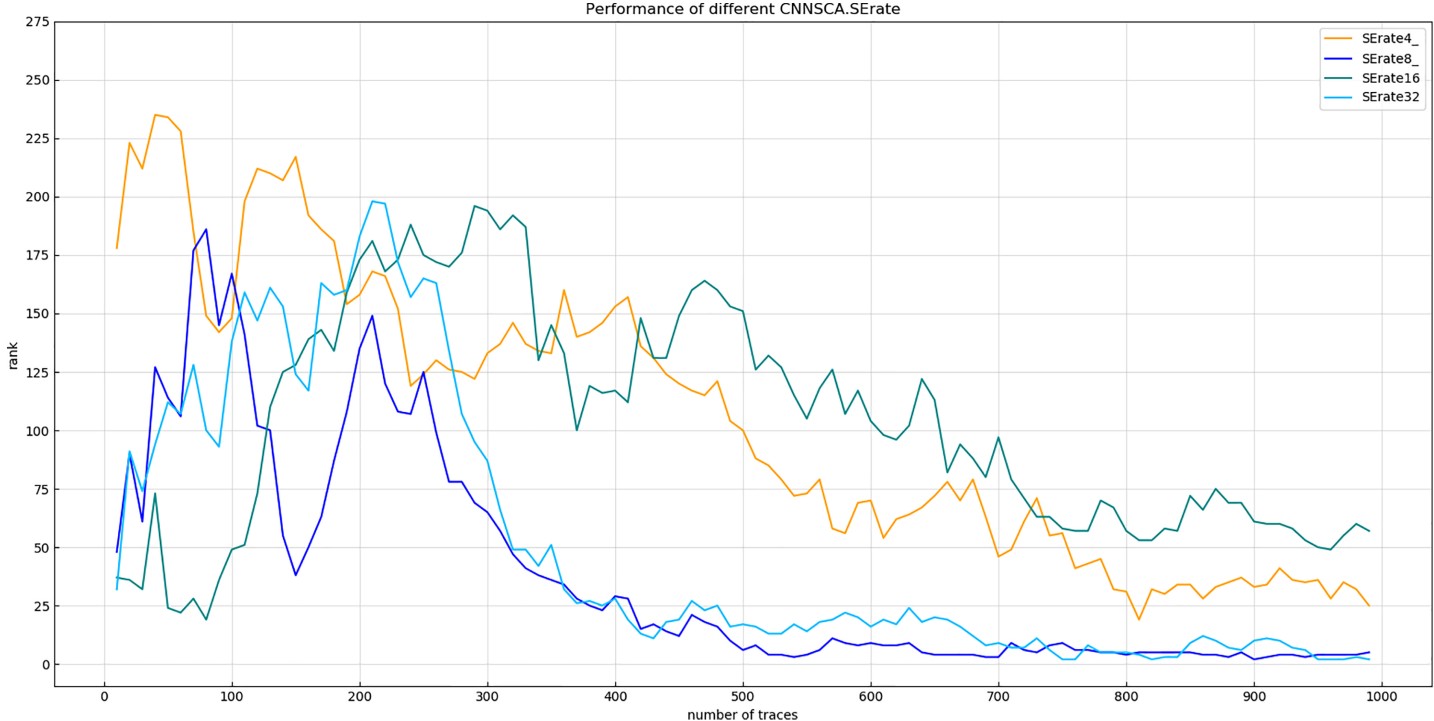

**Figure 12 Convergence of guessing entropy for different SE dimension ratios.** Each curve represents the model guessing entropy of the four SE dimension ratios. The abscissa represents the number of energy trajectories used in the attack, and the ordinate represents the order of guessing entropy.

Experiment 8: Will test the model attack effect of the four cases where the number of channels in the fully connected layer is 4,096, 3,072, 2,048, and 1,024. The other parameters of these test models are the same as CNNSCAnew. The reason why the number of channels is not set lower than 1,024 is that from the convolutional layer to the fully connected layer, if the vector dimension changes sharply, the feature points of the vector are greatly reduced, which will affect the training effect of the model. The result of experiment 8 is shown in Fig. 14 (Convergence of guessing entropy of the four channel number structure of FC layer).

It is found from Fig. 14 that when the number of channels of the FC layer is 1,024, the guessing entropy of its structure converges fastest and continues to be stable. Therefore, 1,024 is selected as the reference for the number of channels in the FC layer of the CNNSCAnew structure.

In summary, the parameter benchmark of the CNNSCAnew structure has been optimized. The new structure parameters are shown in Table 4 (CNNSCAnew Configuration).

Three Selection and optimization of CNN training parameters for side-channel cryptanalysis

Almost all experiments in the Methods 2.2 use the three parameters of 75 iterations, 200 batches of learning, and $1 \times 10^{-4}$ learning rate for experiments. These training parameters

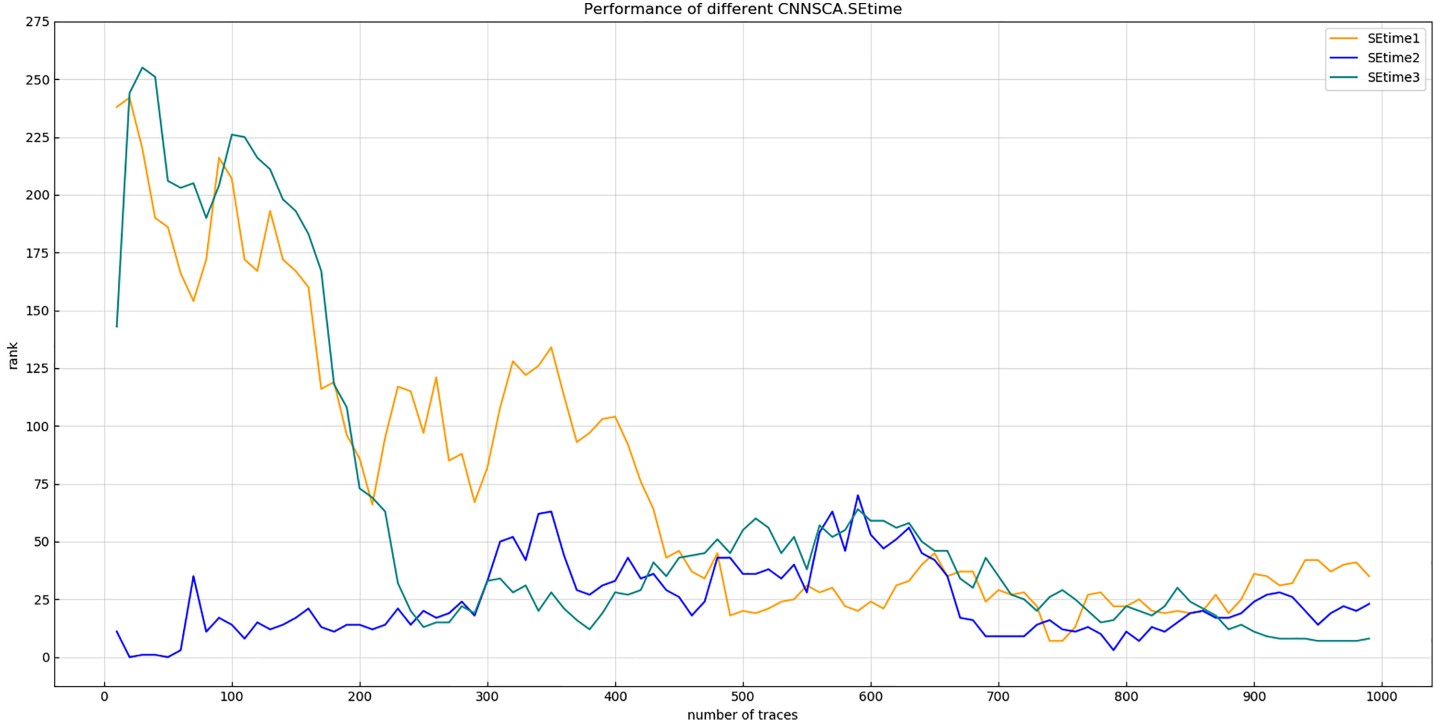

**Figure 13 Convergence of guessing entropy for different number of SE cycles.** Each curve represents the model guessing entropy of the three SE cycles. The abscissa represents the number of energy trajectories used in the attack, and the ordinate represents the order of guessing entropy.

have little effect on the experimental effects of optimizing various structural parameters, but It is not the optimal setting. The convolutional network of deep learning is applied to the side-channel attack, and these training parameters should also be tuned according to the actual processed side-channel signal data. The order of training parameter tuning is usually learning rate, batch learning amount, and number of iterations (*Smith et al., 2017*; *Soltanolkotabi, Javanmard & Lee, 2017*). In the experiment of training parameter optimization, the current CNNSCAnew structure is used. The training parameter optimization experiment process is as follows:

(1) Learning rate

The learning rate is a hyperparameter that is artificially set. The learning rate is used to adjust the size of the weight change, thereby adjusting the speed of model training. The learning rate is generally between 0–1. The learning rate is too large, and the learning will be accelerated in the early stage of model training, making it easier for the model to approach the local or global optimal solution, but there will be large fluctuations in the later stage of the training, and even the value of the loss function may hover around the minimum value, which is always difficult to reach Optimal solution; the learning rate is too small, the model weight adjustment is too slow, and the number of iterations is too much.

Experiment 9: Will test the impact of five commonly used learning rates on the model's cryptanalysis effect, namely $lr1 = 1 \times 10^{-2}$, $lr2 = 1 \times 10^{-3}$, $lr3 = 1 \times 10^{-4}$, $lr4 = 1 \times 10^{-5}$,

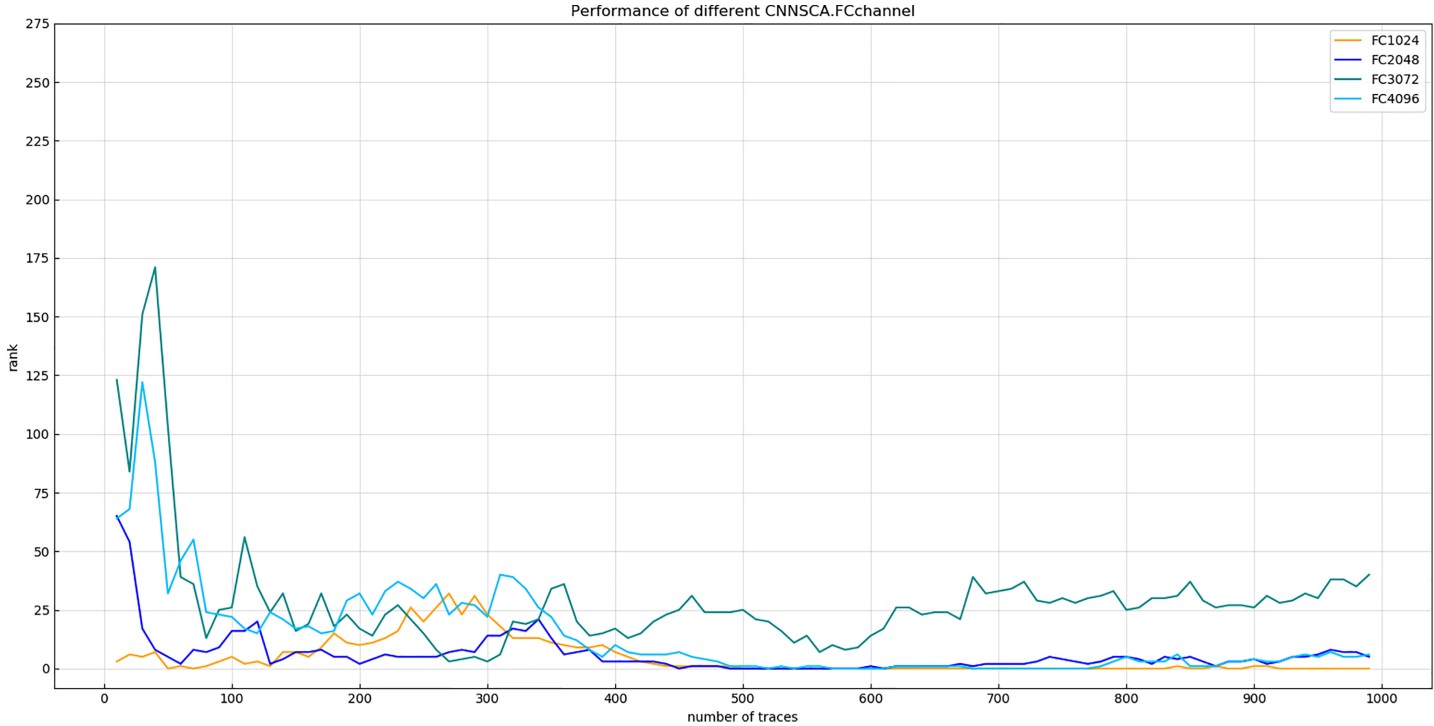

**Figure 14 Convergence of guessing entropy of the four channel number structure of FC layer.** Each curve represents the model guessing entropy of the four FC layer structures. The abscissa represents the number of energy trajectories used in the attack, and the ordinate represents the order of guessing entropy.

**Table 4 CNNSCAnew configuration.**

| ConvNet configuration | | | |
|---|---|---|---|
| Input(1x700 vector) | | | |
| Block1 | (Conv3-32)x1 | Same\ReLU | AveragePool (2,2) |
| Block2 | (Conv3-64)x1 SEx2(1/8) | Same\ReLU | AveragePool (2,2) |
| Block3 | (Conv3-128)x2 SEx2(1/8) | Same\ReLU | AveragePool (2,2) |
| Block4 | (Conv3-256)x2 SEx2(1/8) | Same\ReLU | AveragePool (2,2) |
| Block5 | (Conv3-512)x2 SEx2(1/8) | Same\ReLU | AveragePool (2,2) |
| (FC-1024)x2, ReLU | | | |
| (FC-256)x1, Soft-max | | | |
| Model compile (crossentropy, RMSprop) | | | |

$lr5 = 1 \times 10^{-6}$. The result of experiment 9 is shown in Fig. 15 (Convergence of model guessing entropy under five learning rates).

Figure 15 reflects that when the learning rate is lr2, the guessing entropy of CNNSCAnew converges fastest and is the most stable. Therefore, $1 \times 10^{-3}$ is selected as the learning rate benchmark of the CNNSCAnew structure.

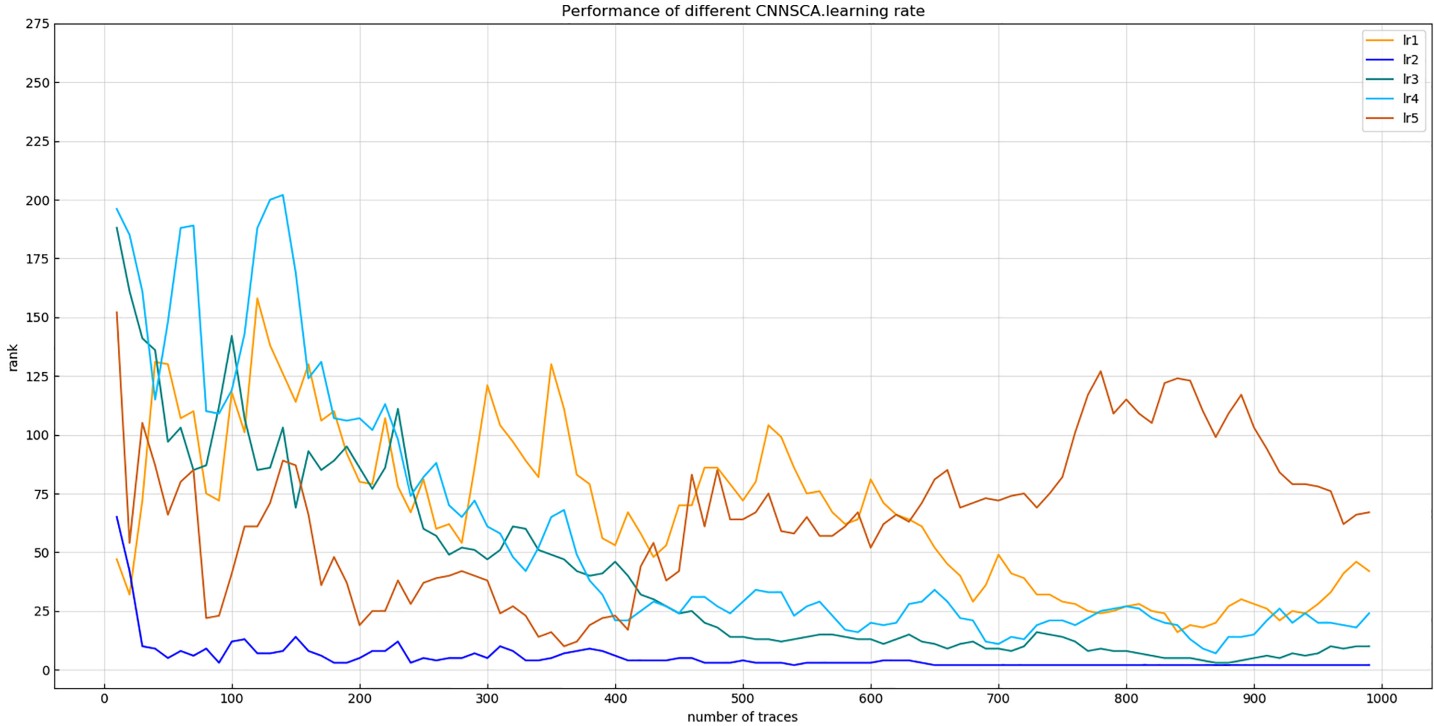

**Figure 15 Convergence of model guessing entropy under five learning rates.** Each curve represents the model guessing entropy of five learning rates. The abscissa represents the number of energy trajectories used in the attack, and the ordinate represents the order of guessing entropy.

**(2) Batch size**

The appropriate batch size is more important for the optimization of the model. This parameter does not need to be fine-tuned, just take a rough number, usually $2^n$ (GPU can play a better performance for batches of the power of 2). A batch size that is too large will be limited by the GPU memory, the calculation speed will be slow, and it cannot increase indefinitely (the training set has 50,000 data); it cannot be too small, which may cause the algorithm to fail to converge.

Experiment 10: According to the size of the ASCAD data set in Section Materials 4, this experiment selects the batch size values: 32, 64, 128, and 256 for the experiment. The result of experiment 10 is shown in Fig. 16 (Convergence of model guessing entropy under four batches).

It can be seen from Fig. 16 that when the batch learning amount is 128, the guessing entropy of CNNSCAnew converges fastest and is the most stable. Therefore, 128 is selected as the batch size benchmark of CNNSCAnew structure.

**(3) Number of iterations (epoch)**

The number of iterations is related to the fitting performance of the CNNSCA model. The model has been fitted (the accuracy rate reaches 1), and there is no need to continue training; on the contrary, if all epochs have been calculated, but the loss value of

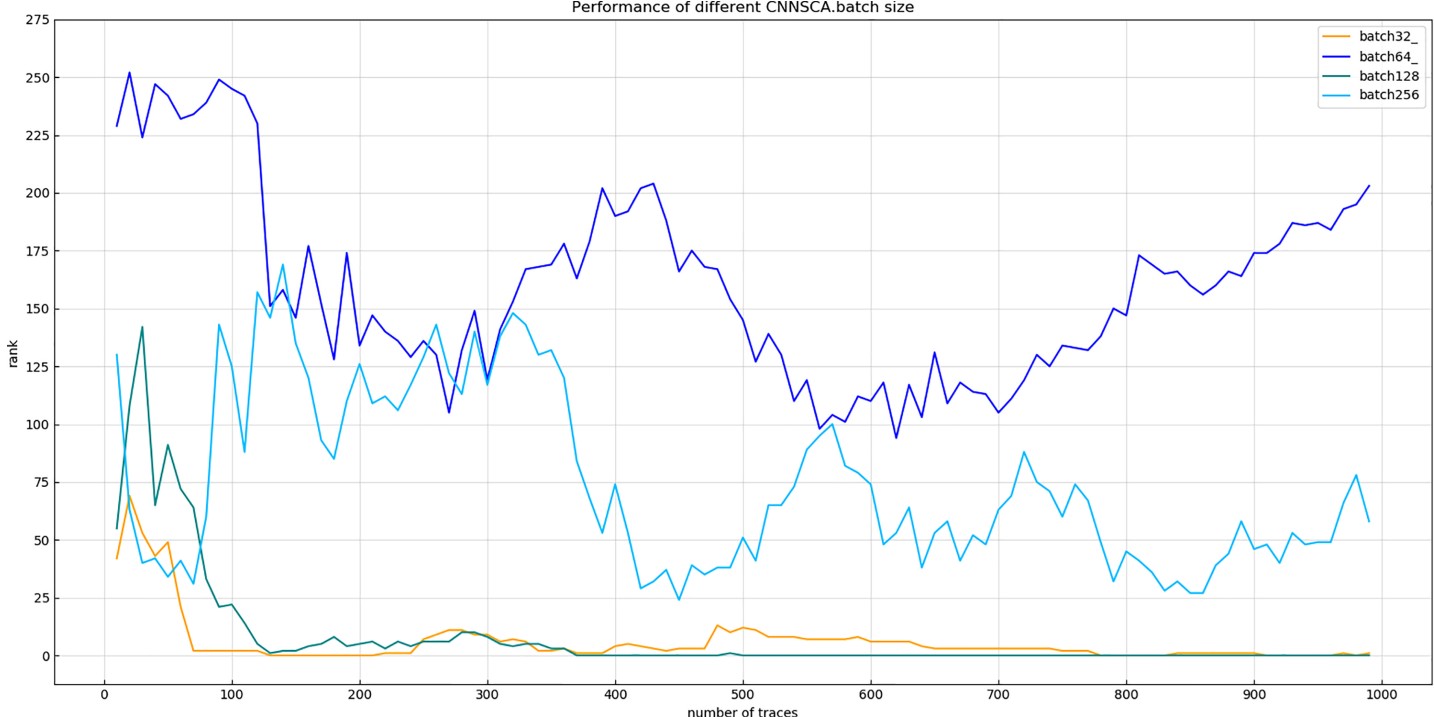

**Figure 16 Convergence of model guessing entropy under four batches.** Each curve represents the model guessing entropy of the four training batches. The abscissa represents the number of energy trajectories used in the attack, and the ordinate represents the order of guessing entropy.

the model is still declining, and the model is still optimizing, then the epoch is too small. Should increase. At the same time, the number of iterations of model training also refers to the actual cryptanalysis effect of the model, which is measured by guessing entropy.

Experiment 11: In experiments 1–10, almost all use iteration number 75 to train the CNNSCA model. This experiment will center on iteration number 75, and train the CNNSCAnew model at 10 intervals in the upper and lower intervals to further optimize the model parameter epoch. The interval number of 10 is chosen because the step interval is too small, and the error loss of model training is not much different, so the setting is meaningless; the interval is too large, and repeated experiments may be required to determine an appropriate number of iterations. Therefore, Experiment 11 will test 8 iteration parameters epoch1=15, epoch2=25, epoch3=35, epoch4=45, epoch5=55, epoch6=65, epoch7=75, epoch8=85. The current CNNSCAnew structure has achieved higher training accuracy and breaking performance, in order to reduce model calculation pressure and calculation time, lower iteration parameters are usually selected when the model performs better. Therefore, the upper limit of the epoch test parameter is set to 85. The result of experiment 11 is shown in Fig. 17 (Convergence of model guessing entropy under eight epochs).

From the results in Fig. 17, it is found that the model of epoch1~4 guesses that the entropy does not converge. Separately recalculate the graph of epoch5~8 model. The result is shown in Fig. 18 (Convergence of model guessing entropy under four epochs). It can be

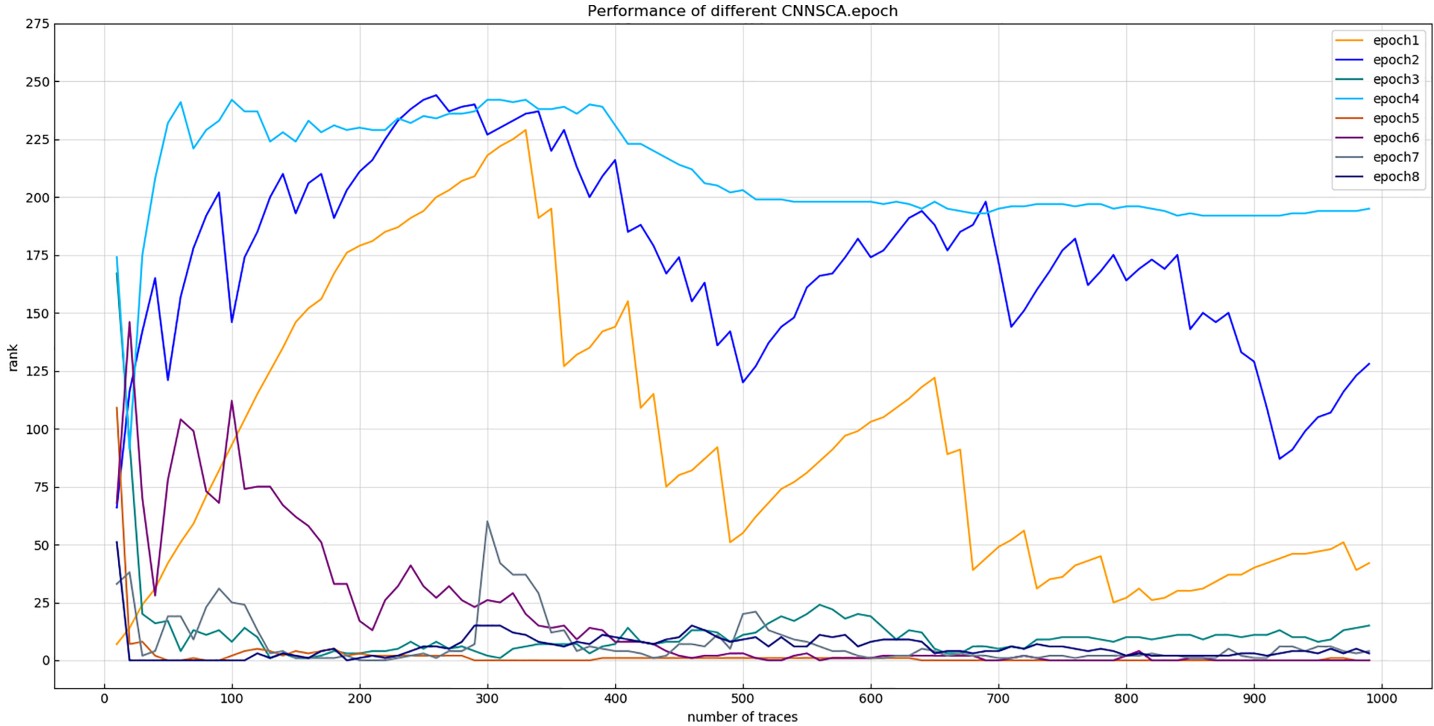

**Figure 17 Convergence of model guessing entropy under eight epochs.** Each curve represents the model guessing entropy of eight training epochs. The abscissa represents the number of energy trajectories used in the attack, and the ordinate represents the order of guessing entropy.

clearly seen that the epoch5–8 model has a convergence trend. Among them, the epoch5 curve is closest to the position of ranking 0, and the epoch8 curve first converges to ranking 0, but afterwards it fluctuates more widely, and it is obviously over-fitting. The convergence of epoch6 and 7 is similar, the curve begins to fluctuate greatly, and it is close to ranking 0 in the later period.

Experiment 12: Continue to debug the epoch parameters in a smaller range, and test the other two iterations with an interval of only 5: epoch60=60, epoch70=70. The trained epoch60 and epoch70 models and the previously trained epoch5, epoch6, and epoch7 models are simultaneously attacked on the target set. The results of Experiment 12 are shown in Fig. 19 (Convergence of model guessing entropy under five epochs).

Figure 19 shows that the guessed entropy of the epoch70 model converges best, and its guessed entropy converges fastest and is the most stable. Therefore, 70 is selected as the training iteration benchmark of the CNNSCAnew structure.

## RESULTS

1 Get a new model CNNSCAnew for attacking ASCAD data set with known first-order mask protection.

According to the 12 sets of experiments in Section Methods 2 and Section Methods 3, the best benchmarks for CNNSCAnew structure parameters and training parameters are

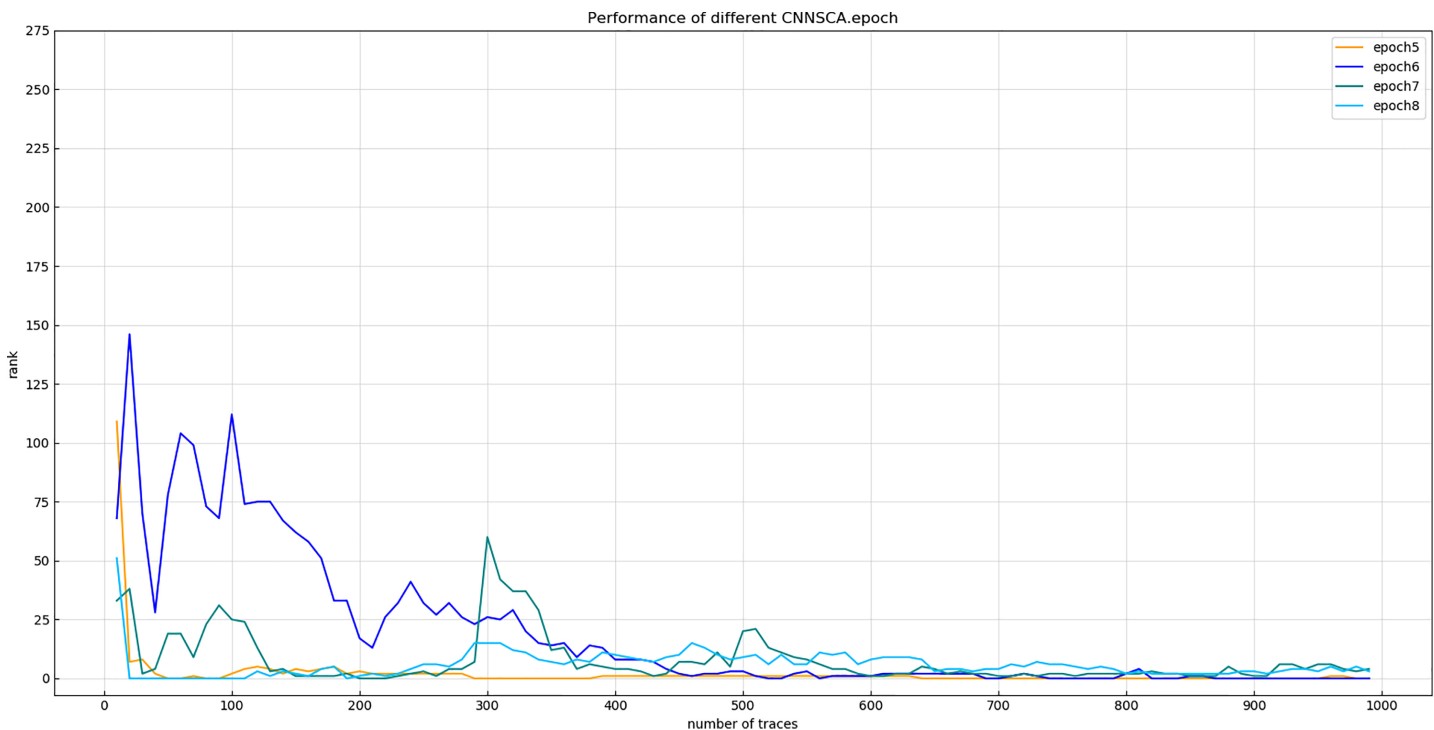

**Figure 18 Convergence of model guessing entropy under four epochs.** Each curve represents the model guessing entropy of four training epochs. The abscissa represents the number of energy trajectories used in the attack, and the ordinate represents the order of guessing entropy.

demonstrated. The CNNSCAnew model contains 5 convolutional blocks, 8 convolutional layers, and 3 fully connected layers. The size of the convolution kernel of each convolution layer is 3, the activation function is ReLU, and the padding is Same. Each convolutional block is equipped with a pooling layer, the pooling layer selects the average pooling mode, and the pooling window is (2,2). The number of output channels of the convolution layer in the convolution block 1–5 starts from 32 and increases by a multiple of 2 in turn. Two SE modules are added after the convolution layer of each convolution block in the convolution block 2–4, and the dimension ratio of the SE is set to 1/8. In the first two fully connected layers, set the number of output channels to 1,024 and the activation function to ReLU. The output channel number of the third fully connected layer is the target classification number 256, and the classification function is Soft-max. The global configuration loss function is crossentropy, the optimization method is RMSprop, the number of training iterations is 70, the learning rate is $1 \times 10^{-3}$, and the batch learning volume is 128. All parameters of the newly obtained CNNSCAnew are shown in Table 5 (CNNSCAnew Configuration).

2 The CNNSCA model design method and the convolutional network hyperparameter optimization scheme for side-channel attack are refined.

The CNNSCA model design method is refined: comprehensively utilize the advantages of VGG-CNNSCA model classification and fitting efficiency and Alex-CNNSCA model

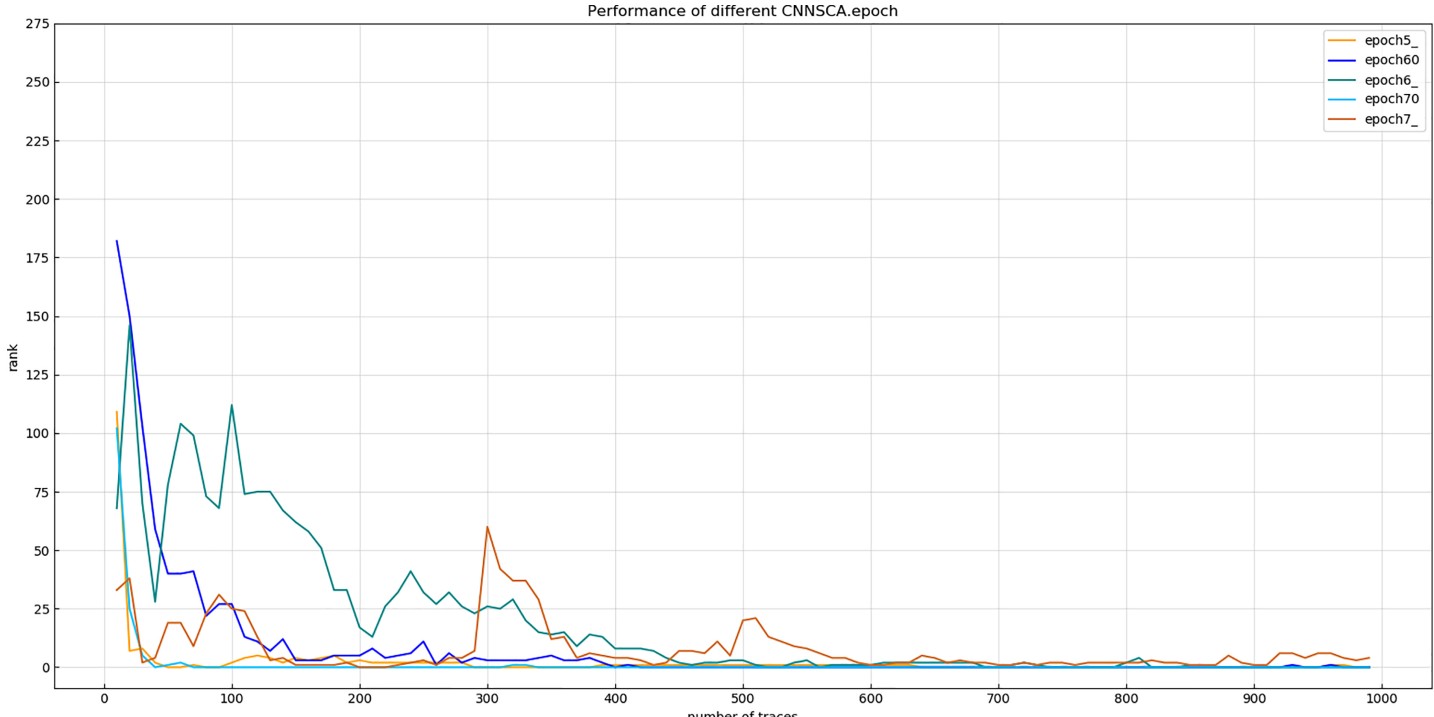

**Figure 19 Convergence of model guessing entropy under five epochs.** Each curve represents the model guessing entropy of five training epochs. The abscissa represents the number of energy trajectories used in the attack, and the ordinate represents the order of guessing entropy.

| Table 5 CNNSCAnew configuration. | | | |
|---|---|---|---|
| **ConvNet configuration** | | | |
| Input(1x700 vector) | | | |
| Block1 | (Conv3-32)x1 | Same\ReLU | AveragePool (2,2) |
| Block2 | (Conv3-64)x1 SEx2(1/8) | Same\ReLU | AveragePool (2,2) |
| Block3 | (Conv3-128)x2 SEx2(1/8) | Same\ReLU | AveragePool (2,2) |
| Block4 | (Conv3-256)x2 SEx2(1/8) | Same\ReLU | AveragePool (2,2) |
| Block5 | (Conv3-512)x2 SEx2(1/8) | Same\ReLU | AveragePool (2,2) |
| (FC-1024)x2, ReLU | | | |
| (FC-256)x1, Soft-max | | | |
| Model compile (crossentropy, RMSprop) | | | |
| Training parameters ($1 \times 10^{-3}$, 128, 70) | | | |

occupy less computing resources, while using SEnet's SE module to reduce the gradient dispersion problem of error back propagation in deep neural networks to save calculation time, a new basic model of CNNSCA was designed, named CNNSCAbase.

At the same time, the hyperparameter optimization scheme of the convolutional network used for side-channel attacks is refined: design the structural parameter optimization experiment and the training parameter optimization experiment, and use

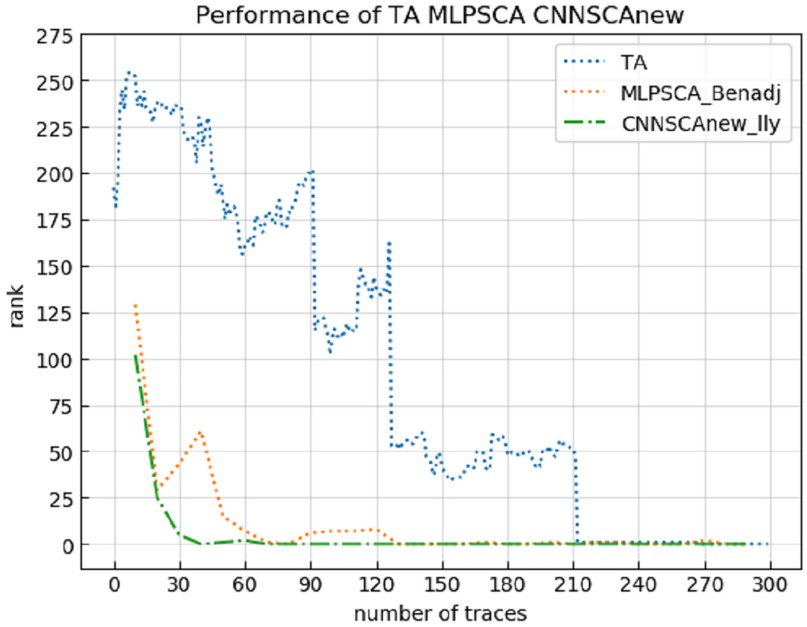

**Figure 20 TA, MLPSCA, CNNSCAnew guessing entropy convergence.** Each curve represents the model guessing entropy of TA, MLPSCA, and CNNSCAnew respectively. The abscissa represents the number of energy trajectories used in the attack, and the ordinate represents the order of guessing entropy.

CNNSCAbase to implement the attack training. According to parameter selection rules, common sense of parameter optimization of CNN model, and data characteristics of actual application scenarios, the test parameters of each experiment are designed, and unnecessary test parameters are excluded. Each time, according to the cryptanalysis results of the experiment, the parameters that make CNNSCAbase's cryptanalysis effect better are selected. Relying on two sets of experimental processes, a hyperparameter optimization scheme is formed, and the hyperparameters finally determined by the experiment are used as the parameters of the new model CNNSCAnew.

## DISCUSSION

Comparative analysis of CNNSCAnew and other profiling side-channel attack methods

1 Comparative analysis of CNNSCAnew, classic template attack and MLPSCA

Experiment 13: Compare the cryptanalysis's performance of CNNSCAnew with the HW-based TA (*Mangard, Oswald & Popp, 2010*) and MLPSCA method proposed by *Benadjila et al. (2018)*. TA and MLPSCA are the profiling methods that performed better in the early traditional profiling methods and the later new profiling methods, respectively. Experiment 13 carried out an attack on the ASCAD data set with a known mask, which represents the realization of the encryption in an unprotected state. The result of experiment 13 is shown in Fig. 20 (TA, MLPSCA, CNNSCAnew guessing entropy convergence).

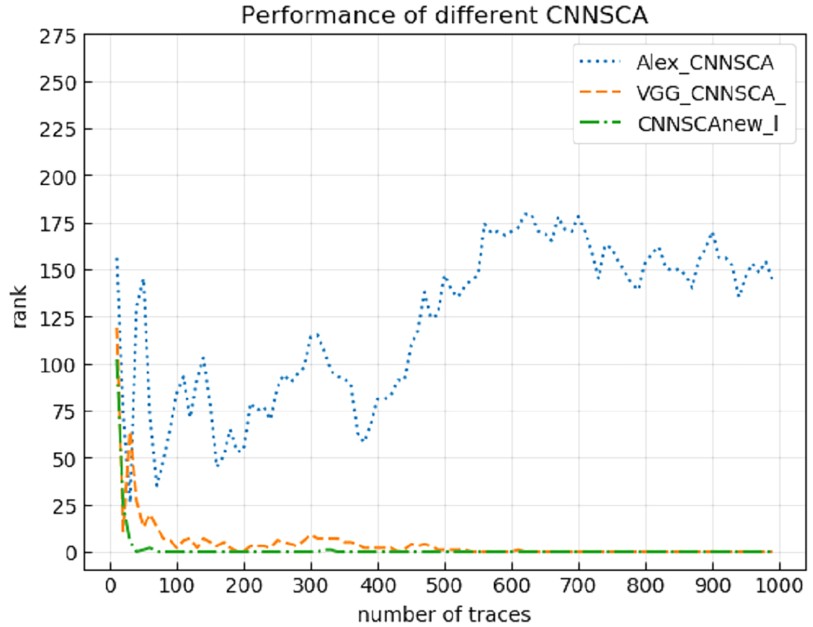

**Figure 21 CNNSCAnew, VGG-CNNSCA, Alex-CNNSCA guessing entropy convergence.** Each curve represents the model guessing entropy of CNNSCAnew, VGG-CNNSCA and Alex-CNNSCA respectively. The abscissa represents the number of energy trajectories used in the attack, and the ordinate represents the order of guessing entropy.

**CNNSCAnew** `Running time:1653.920090106 Seconds`

**VGG-CNNSCA** `Running time:2233.5491708040004 Seconds`

**Figure 22 CNNSCAnew and VGG-CNNSCA training time.**

It can be seen from Fig. 20 that CNNSCAnew's guessing entropy convergence is significantly better than TA and MLPSCA.

2 Comparative analysis with other existing CNNSCA

Experiment 14: Compare the breaking performance of CNNSCAnew model with VGG-CNNSCA (*Benadjila et al., 2018*) and Alex-CNNSCA (*Dongxin et al., 2018*). The latter two methods are the profiling methods with better performance among the latest profiling methods. Among them, VGG-CNNSCA in *Benadjila et al. (2018)* uses the ASCAD public data set, and Alex-CNNSCA in *Dongxin et al. (2018)* uses a self-collected data set. Experiment 14 carried out an attack on the ASCAD data set with a known mask, which represents the realization of the encryption in an unprotected state. The result of experiment 14 is shown in Fig. 21 (CNNSCAnew, VGG-CNNSCA, Alex-CNNSCA guessing entropy convergence).

**Table 6 Comparative analysis of CNNSCAnew, VGG-CNNSCA and Alex-CNNSCA.**

| Three CNNSCA | | CNNSCAnew | VGG-CNNSCA | Alex-CNNSCA |
|---|---|---|---|---|
| CNNSCA Configuration | Convol block1 | (Conv3-32)x1 Same\ReLU AveragePool (2,2) | (Conv11-64)x1 Same\ReLU AveragePool (2,2) | (Conv11-96)x1 Same\ReLU MaxPool (2,2) |
| | Convol block2 | (Conv3-64)x1 SEx2(1/8) Same\ReLU AveragePool (2,2) | (Conv11-128)x1 Same\ReLU AveragePool (2,2) | (Conv5-256)x1 Same\ReLU MaxPool (2,2) |
| | Convol block3 | (Conv3-128)x2 SEx2(1/8) Same\ReLU AveragePool (2,2) | (Conv11-256)x1 Same\ReLU AveragePool (2,2) | (Conv3-384)x1 Same\ReLU |
| | Convol block4 | (Conv3-256)x2 SEx2(1/8) Same\ReLU AveragePool (2,2) | (Conv11-512)x1 Same\ReLU AveragePool (2,2) | (Conv3-384)x1 Same\ReLU |
| | Convol block5 | (Conv3-512)x2 SEx2(1/8) Same\ReLU AveragePool (2,2) | (Conv11-512)x1 Same\ReLU AveragePool (2,2) | (Conv3-256)x1 Same\ReLU MaxPool (3,3) |
| | dense layer | (FC-1024)x2, ReLU (FC-256)x1, Soft-max | (FC-4096)x2, ReLU (FC-256)x1, Soft-max | (FC-4096)x2, ReLU (FC-256)x1, Soft-max |
| | Learning rate | $1 \times 10^{-3}$ | $10^{-5}$ | $10^{-2}$ |
| | Batch size | 128 | 200 | 10 |
| | epoch | 70 | 75 | 20 |
| CNNSCA Performance | calculating time | 28 min | 37 min | 2 h |
| | Guess entropy | 61 | 650 | Did not converge |

It can be seen from Fig. 21 that the CNNSCAnew proposed in this paper has a better guessing entropy convergence than other CNNSCAs. In *Benadjila et al. (2018)*, the guessing entropy of VGG-CNNSCA requires at least 650 power consumption traces to converge to rank zero, and the model training time takes 37 min. The CNNSCAnew method constructed in this paper only requires 61 power consumption traces, and the model training time only needs about 28 min. The training time of CNNSCAnew and VGG-CNNSCA in this paper are shown in Fig. 22 (CNNSCAnew and VGG-CNNSCA training time).

After comparing CNNSCAnew with VGG-CNNSCA and Alex-CNNSCA, the model comparison analysis and the cryptanalysis performance comparison analysis, the results are summarized in Table 6 (Comparative analysis of CNNSCAnew, VGG-CNNSCA and Alex-CNNSCA) to show.

## CONCLUSIONS

Among the profiling side-channel cryptography attack methods, the most popular one is CNNSCA, a side-channel attack method combined with deep learning convolutional

neural network algorithms. Its cryptanalysis performance is significantly better than traditional profiling methods. Among the existing CNNSCA methods, the CNNSCA network models that achieve cryptanalysis mainly include CNNSCA based on the VGG variant (VGG-CNNSCA) and CNNSCA based on the Alexnet variant (Alex-CNNSCA). The learning capabilities and cryptanalysis performance of these CNNSCA models it is not optimal. The paper aims to explore effective methods to obtain the performance gains of the new side-channel attack method CNNSCA.

After studying the related knowledge, necessary structure and core algorithm of CNNSCA, the paper found that CNNSCA model design and hyperparameter optimization can be used to improve the overall performance of CNNSCA. In terms of CNNSCA model design, the advantages of the VGG-CNNSCA model classification and fitting efficiency and the Alex-CNNSCA model occupying less computing resources can be used to design a new CNNSCA basic model. In order to better reduce the gradient dispersion problem of error back propagation in the deep network, it is a very effective method to embed the SE module in this basic model; in terms of the hyperparameter optimization of the CNNSCA model, the above basic model is applied to side-channel leakage A known first-order mask data set in the public database (ASCAD). In this specific application scenario, according to the model design rules and actual experimental results, unnecessary experimental parameters can be excluded to the greatest extent. Various hyperparameters of the model are optimized within the parameter interval to improve the performance of the new CNNSCA, and the final determination benchmark for each hyperparameter is given. Finally, a new CNNSCA model optimized architecture for attacking unprotected encryption devices is obtained—CNNSCAnew. The paper also verified through experimental comparison that CNNSCAnew's cryptanalysis effect is completely superior to traditional profiling methods and the new profiling methods in literature (*Benadjila et al., 2018*; *Dongxin et al., 2018*). In the literature (*Benadjila et al., 2018*; *Dongxin et al., 2018*), the results of CNNSCA's guessing entropy are: convergence to 650 and oscillation. The result of CNNSCAnew's guessing entropy proposed in this paper is to converge to a minimum of 61. Under the same experimental environment and experimental equipment conditions, literature (*Benadjila et al., 2018*) took 40 min from model training to attacking the key, while the total calculation time of CNNSCAnew was shortened to 30 min.

It should be noted that, in practice, the results of each training of the CNNSCAnew model will have a slight deviation. This is a normal phenomenon during neural network training and will not affect the average performance of the model. While proposing the new CNNSCA method, the paper also provides a more comprehensive and detailed design plan and optimization method for the side-channel cryptanalysis researchers who need to design the CNNSCA model. In the future, we can use these design schemes and optimization methods to continue to explore the CNNSCA model that is more suitable for attacking protected equipment to achieve efficient attacks on encrypted equipment with protection, which is of great significance to information security and encryption protection.

## ACKNOWLEDGEMENTS

In the paper, from the topic selection of the paper, the structure of the chapter to the scrutiny of words, I got the careful guidance of my tutor, professor Chen Kaiyan; in life, my teacher is very concerned and caring for me. In addition, I sincerely thank professor Li Xiongwei and Zhang Yang. They put forward many opinions and suggestions on my paper, which greatly inspired the students and opened up a lot of thinking. Thanks to the classmates and friends around me. They gave me encouragement and assistance, and let me feel the warmth of this big family. May our friendship last forever. Thanks to the university for providing me with a good learning platform, which gave me a new start.

### Funding

The authors received no funding for this work.

### Competing Interests

The authors declare that they have no competing interests.

### Author Contributions

- Yun Lin Liu conceived and designed the experiments, performed the experiments, analyzed the data, performed the computation work, prepared figures and/or tables, authored or reviewed drafts of the paper, and approved the final draft.
- Yan Kai Chen analyzed the data, authored or reviewed drafts of the paper, and approved the final draft.
- Wei Xiong Li analyzed the data, authored or reviewed drafts of the paper, and approved the final draft.
- Yang Zhang analyzed the data, authored or reviewed drafts of the paper, and approved the final draft.

### Data Availability

The power consumption data set representing the unprotected encryption device in the ASCAD database and the main code of the CNN model of side-channel cryptanalysis proposed in the article are available in the Supplemental Files.

### Supplemental Information

Supplemental information for this article can be found online at http://dx.doi.org/10.7717/peerj-cs.829#supplemental-information.

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
