# Peer review of "Model design and parameter optimization of CNN for side-channel cryptanalysis"

_PeerJ Computer Science, doi:10.7717/peerj-cs.829_

## Round 0.1 · original submission · Major Revisions

I have received detailed reports about the paper, which should be very useful for improving your paper. Please provide point by point responses with your revision. Thanks.

Reviewer 1 ·

Basic reporting

The overall passage is well structured with adequate deep learning background information, which I think provides basic intuitions for researchers who want to study deep learning based side channel attack. It should be better if more background information about side channel attack is included, e.g. the meaning of the rank, how it is calculated, the relationship between rank and classification results, and so on.

The language is good, easy to understand. Although there are some typos, which I point below. No problem to use the public database ASCAD, I am not sure it is necessary to publish the author’s training and testing codes, but they share the model structure, which I think is enough.

As for the figures, please rename the figures to make the indexes consistent with the figure titles. Now they are confusing.

Typos:
In line 53, secret.key. ->secret key.
In line 101, deep network The problem-> deep network. The problem
In line 108, Parameter -> parameter
In line 122, please rephrase your sentence
In lines 282-283, x_j is not the inputs of all neurons but one neuron
In line 395, please rephrase your sentence
Please check the name ‘Conv’, there are many ‘Cnov’ in the passage

Experimental design

I like the way the authors design the experiment, the whole experiments are designed in sequential way, and the CNNSCAnew is updated after every experiment. They analyze the model parameters first and then switch to the global parameters, and finally determine the optimal model structure and parameters.

However, I think the authors miss one important thing. They should explain why the whole experiments flow is designed like this. For example, why they choose to determine the model parameters first instead of the global training parameters? and in section 2.2, why they choose to determine the number of Conv layers first instead of kernel size or the number of filters first?

In addition, I know there are lots of parameters to be optimized and it is tough to go through all the combinations. So I am thinking that whether the impacts of different parameters on the model performance are independent, if yes, there is no problem to optimize the model parameter by parameter, if no, then the whole experiment should be carefully addressed in order to find the optimal model. I will appreciate it if the authors can give more details about this.

Finally, please explain why choose the epoch number 75 for most of the experiments ? Have you tried other epoch numbers? And I think the authors should also add comparison results between models with SEnet and without SEnet if they want to show SEnet can reduce the gradient dispersion.

Validity of the findings

Based on the conclusions, the authors provide us with a new CNNSCA structure that has better classification results and lower computational time than previous model structures (Benadj, Alex VGG). This part is strongly related to the experiment design part, other than the updated model structure and short training time, I expect more conclusions here.

Additional comments

Please gives more background information about why it is important to use deep learning in side channel attacks and comparison between deep learning performance and traditional side channel attack analysis methods.

Reviewer 2 ·

Basic reporting

- This paper proposes a convolutional neural network-based side-channel attack model design and hyperparameter optimization to improve side-channel attack performance. The paper is mostly clear but there are numerous grammatical errors. For example, the sentences in lines 90, 124, 319-323, 339, and 639 need revision.
- The paper contains sufficient references. I can only recommend the authors add a reference for the MNIST dataset in line 306 of the paper.
- Table 1 is not clearly explained in the text. The authors can mention one or two sentences about remarks in table 1.
- Figure 1 is not explained in detail.
- For figure 2, the authors can use labels to show which data is input, kernel, and output.
- In line 135, "BP algorithm" is mentioned but readers may not know the BP. What does BP stand for?

Experimental design

- The performance of convolutional-based side-channel attacks has dramatically improved by researchers over the last 5 years. However, there is still room to develop it.
- In this paper, there are valuable experimental results. The authors compare the performance of the proposed model with other models in the literature. The proposed method is superior to the other methods.

Validity of the findings

- Although the authors did not propose a completely new method/model, they adapted existed methods well to generate a new model. This combined model is not proposed previously in the literature. The experimental results show that it is better than the other methods.

Reviewer 3 ·

Basic reporting

This paper aims to improve the state of art of deep learning based side channel-attacks by proposing a new model, named CNNSCAnew. The architecture of this model relies on a new kind of block of layers, the SE module, whose effectiveness has already been established in the domain of computer vision. A fine tunning of the hyperparameters is performed by studying the impact of each parameters independently in several experiences. The performance of the final model is compared with the state of the art models on the open dataset ASCAD.

From the editorial side, this paper is not well written. There is a lot of approximations, confusions and contradictions, mostly due to the fact that some english technical terms are not correctly used. Some sentences are not grammatically correct. The overall meaning of the text can be understood (or somehow guessed), but it is very tedious and some typos or formulations can be easily corrected with a more careful review of the paper.

Here are some examples found in the abstract and introduction (the list is not exhaustive)
l12. decryption -> cryptanalysis
l13. decryption -> cryptanalysis
l13. Alex -> AlexNet
l14. there is Model training -> the trained model
l23. the SEnet module : the "SEnet" abbreviation is not yet defined
l24. unnecessary parameters are maximized: do you mean "optimized" instead of maximized?
l28. Optimize ... CNSSCA : the meaning of this sentence is not clear, do you mean : "We optimized the various ... CNNSCA"?
l26. first-order mask data set of -> first-order masked dataset from
l34. guess entropy -> guessing entropy
l35. successful attack key -> successful recovery of the key
l36. the performance was better -> we obtained better performance
l42. and using cryptographic algorithms to leak ... harware encryption: the formulation is not correct, I would suggest something like : "by using the leakage of cryptographic algorithms during the computation of data (...) on hardware devices"
l43. radiation Etc -> radiation, etc.
l44. to crack : the word is a little bit slang
l50: the weak -> weak
l53: the secret. key: typo
l54. the best decryption effect -> the best cryptanalysis attack
l58. energy traces -> power consumption traces
l69. At present, at home and abroad : ?
l79. the best perfomance of breaking secrets -> the best cryptanalysis performance
l123. declassification -> classification

Experimental design

The study of new attention mechanisms for side-channel attacks is in line with the current trends in deep learning research and the efforts to use open dataset and reproductible results are appreciated. The results are interesting and the overall methodology for fine tunning the deeplearning model sounds correct. All the model parameter choices are published and the code is available, which is very valuable to the side channel community. The experiments rely on an open dataset and they are reproductible, which strenghten the confidence in the results. The hyperparameters are carefully selected and their impact are thorougly studied. The figures depicting the results are clear. The final model, named CNNSCAnew, contributes to improve the state of the art on SCA attacks based on deeplearning.

Validity of the findings

Nevertheless, due to the poor level of english and typos, it is difficult to undestand the description of the experiments and sometimes the results are not clear. By example, in the section "Structural parameter optimization", experiment 1, the authors have tested three deeplearning architectures with different number of convolutional layers for each block (namely 1, 2 and 2/3). At line 416, they claim that "if [the number of layers] exceeds 2 or more, ..., the 8G[bytes] GPU memory ... will be exhausted, unable to run code". But they computed the guessing entropy for some of their architecture where the number of layer was 2 (cnov2 and cnov3), which is in contradiction with the fact that they were unable to run the code.

Moreover some claims are vague or not correct. By example l62. "but it also loses effectiveness when attacking encryption with protection." : this statement is not true : in the ASCAD paper, a MLP is successfuly applied to a protected AES encrytion. Another example at l539, the authors claim that if the number of channels of the dense layer is low, then the computational complexity increase. However if n is the input dimension and c the number of channel, then the number of computation is equal to c*n and it increases with the number of channel.

Finally, the comparaison with the state of the art models lacks some robustness. I recommand to perform a 10-fold crossvalidation or a validation method with a high number of different training/testing steps: the results obtained with these well-known methods will be more significant than a single training/testing evaluation and will reduce bias.

---

## Round 0.2 · accepted · Accept

The paper has been well revised. I believe all questions raised in the first round of review have been addressed. I am happy to recommend it for publication.